# Nanobody-thioesterase chimeras to specifically target protein palmitoylation

Chien-Wen Kuo[1], Caglar Gök[1,2], Hannah Fulton[1], Eleanor Dickson-Murray[1], Samuel Adu[1], Emily K. Gallen[1,3], Sheon Mary[1], Alan D. Robertson[1], Fiona Jordan[1], Emma Dunning[1], William Mullen[1], Godfrey L. Smith[1] & William Fuller[1] ✉

The complexity of the cellular proteome is massively expanded by a repertoire of chemically distinct reversible post-translational modifications (PTMs) that control protein localisation, interactions, and function. The temporal and spatial control of these PTMs is central to organism physiology, and mis-regulation of PTMs is a hallmark of many diseases. Here we present an approach to manipulate PTMs on target proteins using nanobodies fused to enzymes that control these PTMs. Anti-GFP nanobodies fused to thioesterases (which depalmitoylate protein cysteines) depalmitoylate GFP tagged substrates. A chemogenetic approach to enhance nanobody affinity for its target enables temporal control of target depalmitoylation. Using a thioesterase fused to a nanobody directed against the Ca(v)1.2 beta subunit we reduce palmitoylation of the Ca(v)1.2 alpha subunit, modifying the channel's voltage dependence and arrhythmia susceptibility in stem cell derived cardiac myocytes. We conclude that nanobody enzyme chimeras represent an approach to specifically manipulate PTMs, with applications in both the laboratory and the clinic.

Post-translational modifications (PTMs) of cellular proteins shape numerous aspects of function during a protein's lifetime in a cell. Some PTMs are irreversible steps in protein maturation, but most reversibly tune protein structure and consequently function, and occur in a regulated fashion in response to activation or inhibition of cellular signalling pathways. Mis-regulation of PTMs can be both the cause and effect of numerous human pathologies[1–4].

An enormous body of work has gone into understanding the importance of individual PTMs in both health and disease. Typical approaches to interrogate PTM function involve conservative mutation of candidate amino acids to structurally similar residues that cannot be modified (e.g. cysteine or serine to alanine, tyrosine to phenylalanine), or broad inhibition of cellular pathways with pharmacological agents (e.g. kinase or phosphatase inhibitors). However, such approaches are prone to experimental artefacts. For

example, multiple PTMs can compete for the same amino acid. Cysteines can be palmitoylated, glutathionylated, nitrosylated, or reversibly or irreversibly oxidised. Serines can be phosphorylated or O-GlcNAcylated. In addition, unexpected off- or on-target effects of drugs used to manipulate pathways or enzymes responsible for PTMs can confound experiments. Alternative approaches to interrogate PTM function such as PTM mimetic mutations (e.g. serine to aspartate for phosphorylation[5], cysteine to tryptophan for lipidation[6]) do not fully recapitulate the chemical features of individual PTMs. The use of semi-synthetic post-translationally modified proteins (for example refs. 7–9) avoids this issue but does not generally allow interrogation of protein function in the native cellular environment. On the other hand, incorporating un-natural amino acids into a target protein can facilitate site-specific chemical modification of a single protein in intact cells (for example ref. 10),

[1]School of Cardiovascular & Metabolic Health, College of Medical Veterinary and Life Sciences, University of Glasgow, Glasgow, UK. [2]Present address: School of Natural Sciences, College of Health and Science, University of Lincoln, Lincoln, UK. [3]Present address: Harvard Medical School, Boston, MA, USA. ✉e-mail: Will.Fuller@glasgow.ac.uk

but does not offer the opportunity to evaluate the impact of dynamic PTM cycling on protein function.

A recurring theme in the control of many PTMs is how fidelity and specificity are encoded into the system. Protein kinases, for example, are directed towards their substrates by anchoring proteins that ensure the colocalization of kinases with substrates[11], as well as recruiting other signalling molecules such as phosphatases and phosphodiesterases[12]. zDHHC-containing protein acyl-transferase enzymes, which reversibly palmitoylate protein cysteines, recruit substrates through specific binding sites that are distant from the active site[13,14]. Hence, although the primary and secondary sequence determinants of these PTMs are very different, there is clear commonality in how they are regulated by specific protein interactions.

Single chain antibodies from camelids are small, highly stable entities that bind their target protein with high affinity. Unlike other mammalian antibodies the protein binding unit of camelid antibodies (VHH domain, also known as a nanobody) folds correctly in the mammalian cytoplasm. Given the established importance of protein-protein interactions for controlling PTMs, we hypothesised that fusing nanobodies to enzymes that mediate these PTMs would deliver these enzymes to the nanobody's target protein. In this investigation we validate this approach using an anti-eGFP nanobody to deliver a protein thioesterase to depalmitoylate eGFP tagged proteins. Using a chemogenetic approach to control nanobody affinity for eGFP we achieve temporal control of nanobody binding and protein depalmitoylation. A chimera between a nanobody directed against the voltage sensitive calcium channel Ca(v)1.2 and a thioesterase facilitates Ca(v)1.2 depalmitoylation, tuning the biophysical properties of the channel and preventing arrhythmias in cardiac myocytes. Overall, this work demonstrates the feasibility of targeting post-translational regulation of individual proteins using nanobodies, with broad applications in biology and medicine.

## Results

### A GFP binding nanobody fused to a thioesterase depalmitoyates GFP-tagged substrates

Palmitoylation by zDHHC-PATs reversibly anchors proteins to membranes by conjugating a saturated fatty acid (most commonly C16 palmitate, molecular mass 256 Da) to the amino acid cysteine via a thioester bond. We selected the eGFP binding nanobody LaG-16 to target protein palmitoylation because it is well characterised, with a sub-nanomolar binding affinity, and is established to bind to the common eGFP variants eCFP and eYFP[15]. We first evaluated whether we could target palmitoylation of protein cysteines by fusing LaG-16 to either the amino or carboxyl terminus of the depalmitoylating enzymes APT1 or APT2 (Fig. 1A). Both these enzymes require palmitoylation at their amino termini to anchor them to membranes so they can engage with substrates[16]. We reasoned that fusing the nanobody to the amino termini would interfere with, but might directly substitute for, this targeting. Therefore, nanobody fusions to the amino terminus of the thioesterases (LaG-16-APTx) did not include a flexible linker between the two proteins. We included a flexible linker when fusing LaG-16 to the carboxyl termini of thioesterases (APTx-LaG-16) because we reasoned that an N terminal membrane anchor (palmitate) and a C terminal substrate anchor (the nanobody) might sterically constrain the enzyme without this flexibility.

We used the dually palmitoylated intracellular I-II linker of the α1C subunit of Ca(v)1.2[17] (α1C-IL) as a model substrate to evaluate nanobody efficacy. Neither APT1 nor APT2 are found in the Ca(v)1.2 microdomain in ventricular muscle[18]. Studies using APT1 and APT2 inhibitors in isolated rabbit ventricular myocytes indicate Ca(v)1.2 α1C subunit is not a substrate of either enzyme (Supplementary Fig. 1). We co-expressed eYFP-tagged α1C-IL (YFP-α1C-IL) with thioesterase-nanobody fusions in HEK cells and measured its palmitoylation status using resin-assisted capture. This technique captures palmitoylated

proteins on cysteine-reactive beads and consequently cannot distinguish between singly and doubly palmitoylated species. Hence a reduction in the amount of a dually palmitoylated protein captured indicates this protein has been entirely depalmitoylated. In these experiments all four nanobody constructs acted as powerful depalmitoylating agents for YFP-α1C-IL but did not alter the palmitoylation status of the housekeeping protein flotillin 2 (Fig. 1B). Nanobody fusions to thioesterase carboxyl termini were marginally more potent than fusions at the amino termini. APT1 and APT2 were palmitoylated only when LaG-16 was fused to their carboxyl termini (Fig. 1B). There was no quantitative difference between the extent of depalmitoylation mediated by APT1 or APT2 delivered by LaG-16. We conclude from these experiments that simply recruiting a thioesterase to the locality of a protein is sufficient to induce that protein's depalmitoylation, regardless of whether this protein is usually depalmitoylated by this thioesterase.

The lack of palmitoylation of YFP-α1C-IL in the presence of LaG-16-thioesterase fusions could conceivably result from nanobody engagement blocking access for a palmitoylating enzyme. To control for this possibility, we mutated the catalytic serine of APT1 or APT2 in all nanobody fusions and co-expressed LaG-16 fused to inactive thioesterases with YFP-α1C-IL (Fig. 1C). YFP-α1C-IL remained palmitoylated in the presence of all nanobodies, confirming the requirement for thioesterase activity for YFP-α1C-IL depalmitoylation.

We next sought evidence that LaG-16 engagement with eYFP was required for nanobody-mediated YFP-α1C-IL depalmitoylation. Nanobodies engage their targets via a paratope composed of three complementarity-determining regions (CDRs). Guided by a co-crystal structure of eGFP and LaG-16[19] we mutated one amino acid in each CDR to a bulky tryptophan to generate a triple mutant F30W/ T55W/G103W LaG-16 which we hypothesised would have reduced affinity for eGFP (Fig. 1D). Co-immunoprecipitation experiments confirmed reduced affinity of this '3W' mutant to YFP-α1C-IL (Fig. 1E). APT1 and APT2 fusions to 3W-LaG-16 displayed significantly blunted depalmitoylating activity towards YFP-α1C-IL (Fig. 1F). These experiments confirm the importance of nanobody engagement for target depalmitoylation.

### Investigating the broad applicability of nanobody-mediated depalmitoylation

Having validated the specificity of our depalmitoylating nanobody reagents towards YFP-α1C-IL we evaluated their impact on palmitoylation of fluorescently tagged integral membrane proteins that principally reside at the cell surface. We first evaluated proteins established to undergo dynamic cycles of palmitoylation and depalmitoylation. The type 1 membrane protein phospholemman (PLM) regulates the sarcolemmal Na/K ATPase in the heart[20,21]. PLM is palmitoylated at two juxtamembrane cysteines by zDHHC5 and depalmitoylated by the antioxidant protein peroxiredoxin 6[14,22–24]. APT2-LaG-16 but not APT1-LaG-16 depalmitoylated PLM-eYFP when they were co-expressed in HEK cells (Fig. 2A). The cardiac sodium/calcium exchanger NCX1 is palmitoylated at a single cysteine in its large regulatory intracellular loop[25], which is depalmitoylated by APT1[26,27]. Co-expression with APT2-LaG-16 did not alter the palmitoylation status of full length NCX1 with eYFP fused in its intracellular loop (Fig. 2B). APT1 fused to 3W-LaG-16 modestly depalmitoylated eYFP tagged NCX1, much as would be expected for an APT1 substrate. APT1 fused to LaG-16 potently depalmitoylated eYFP tagged NCX1.

Next, we evaluated whether stably palmitoylated proteins which do not undergo cycles of palmitoylation and depalmitoylation could be depalmitoylated using thioesterase-nanobody chimeras. Caveolin isoforms are irreversibly palmitoylated early in the biosynthetic pathway[28]. Caveolin-3, the muscle specific isoform, is palmitoylated at 6 cysteine residues, with Cys98 the predominant site of palmitoylation[29]. We fused eYFP to the amino terminus of caveolin-3

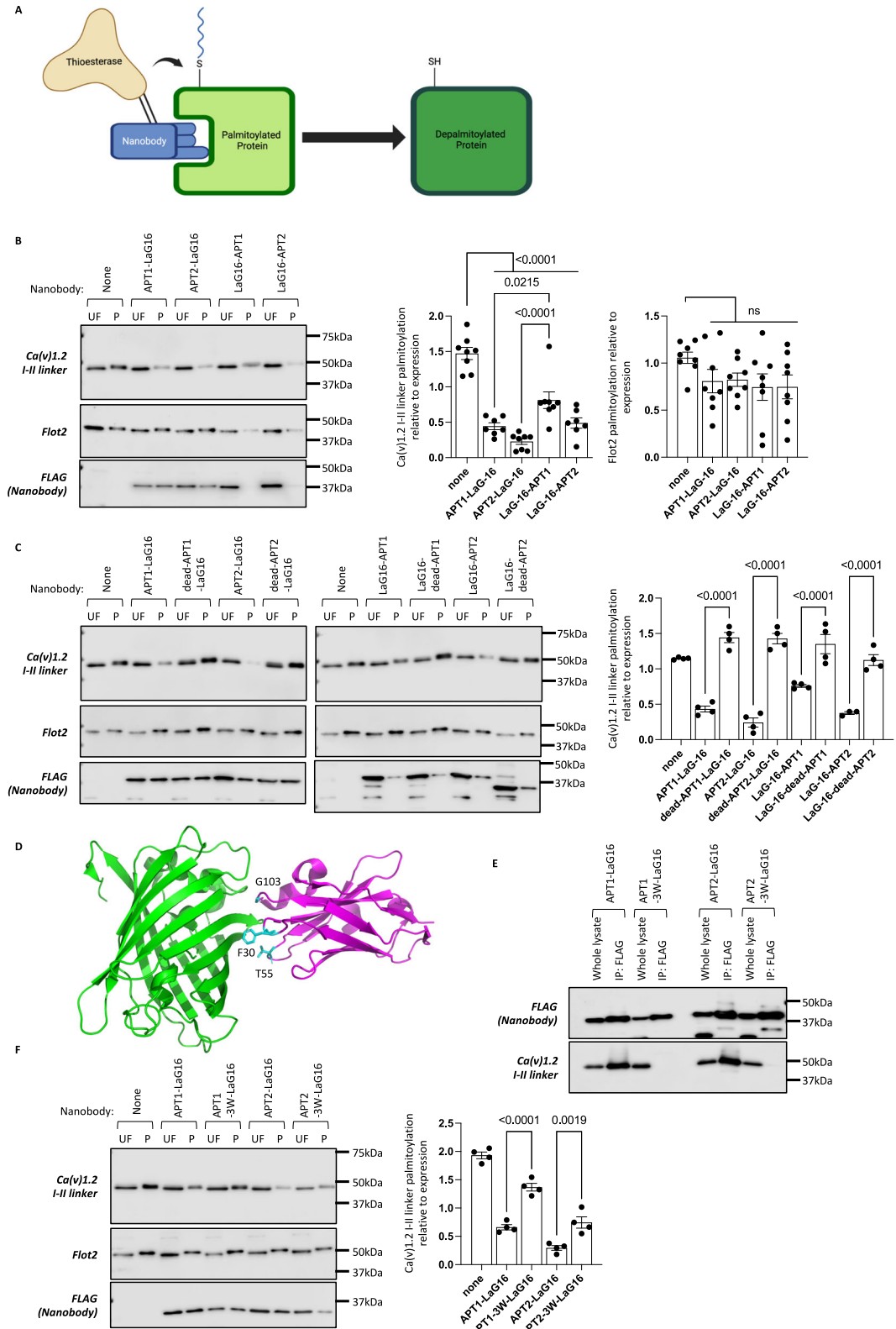

**Fig. 1 | Nanobodies targeting protein palmitoylation. A** Schematic. Nanobody binding to its palmitoylated target protein delivers a thioesterase which removes the 16 carbon fatty acid palmitate (depicted in blue) from the target (Created in BioRender. Fulton, H. (2025) https://BioRender.com/n03e444). **B** Impact of anti-GFP nanobody LaG16 fused to the amino or carboxyl terminus of thioesterases (APT1, APT2) on YFP-Ca(v)1.2 I-II-linker palmitoylation. UF: unfractionated cell lysate; P: purified palmitoylated proteins; Flot2: flotillin-2. Data are means ± SEM from $n = 8$ independent experiments. **C** Impact of LaG16 fused to catalytically inactive ('dead') APTs on YFP-Ca(v)1.2 I-II-linker palmitoylation. Data are means ±

SEM from $n = 4$ independent experiments. **D** Structure (PDB: 6LR7) of eGFP (green) in complex with LaG16 (magenta) highlighting the positions of F30, T55 and G103 (cyan) in CDRs 1, 2 and 3. **E** Wild-type and mutated ('3W') LaG16 binding to YFP-Ca(v)1.2 I-II-linker assessed by immunoprecipitating the FLAG tagged nanobody. This experiment was repeated independently three times with similar results. **F** Impact of LaG16 and mutated LaG16 fused to APTs on YFP-Ca(v)1.2 I-II-linker palmitoylation. Data are means ± SEM from $n = 4$ independent experiments. Statistical comparisons: one-way ANOVA followed by Tukey's multiple comparisons test. Source data are provided as a Source Data file.

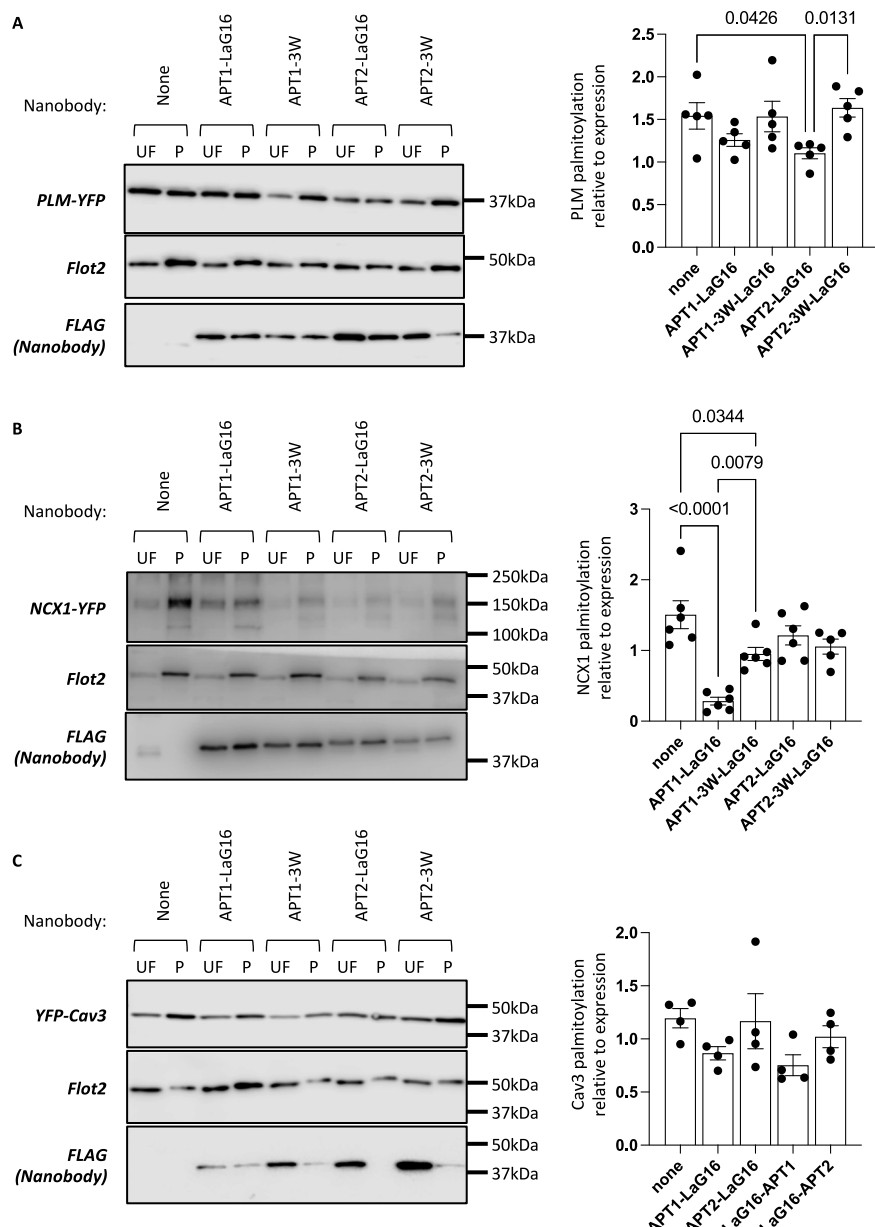

**Fig. 2 | Nanobodies targeting protein palmitoylation: impact on integral membrane protein palmitoylation.** Impact of anti-GFP nanobody LaG16 or non-binding nanobodies (3W) fused to depalmitoylating enzymes (APT1, APT2) on palmitoylation of YFP-tagged PLM (**A**, Data are means ± SEM from n = 5 independent experiments), YFP-tagged NCX1 (**B**, Data are means ± SEM from n = 6 independent experiments), and YFP caveolin-3 (**C**, Data are means ± SEM from n = 4 independent experiments). Statistical comparisons: one-way ANOVA followed by Tukey's multiple comparisons test. Source data are provided as a Source Data file.

with all cysteines apart from Cys98 mutated to alanine and evaluated whether this single palmitoylation site could be targeted by co-expressing APT1 or APT2 fused to LaG-16. Neither reagent was capable of depalmitoylating eYFP-caveolin-3 (Fig. 2C). We conclude from these experiments that stably palmitoylated proteins cannot be depalmitoylated using thioesterase-nanobody chimeras.

Collectively these results confirm the broad applicability of thioesterase-nanobody chimeras as selective depalmitoylating agents towards dynamically, but not constitutively palmitoylated proteins.

### Chemogenetic control of protein palmitoylation

We set out to achieve temporal control of nanobody-mediated depalmitoylation. Inserting circularly permutated bacterial dihydrofolate reductase (cpDHFR) into CDR3 of the GFP enhancer nanobody generates a ligand-modulated antibody fragment (LAMA)[30]. cpDHFR does not impact nanobody engagement with GFP unless both the DHFR cofactor NADPH and the DHFR inhibitor trimethoprim (TMP) are present (Fig. 3A). In intact cells, where NADPH levels are typically $3\,\mu M^{31}$, the change in cpDHFR conformation upon TMP addition causes the nanobody to release its target protein[30]. We created APT2 fusions to two different LAMAs, in which cpDHFR was inserted after either G97 or F98 in CDR3 of the GFP enhancer nanobody, and co-expressed them with YFP tagged Ca(v)1.2 α1C-IL in HEK cells in the presence and absence of TMP. Treating cells with TMP overnight reduced co-immunoprecipitation of APT2-LAMA-F98 and abolished co-immunoprecipitation of APT2-LAMA-G97 with α1C-IL (Fig. 3B). In the absence of TMP both LAMAs potently depalmitoylated α1C-IL. Overnight application of TMP did not influence depalmitoylation by APT2-LAMA-F98, but entirely abolished the ability of APT2-LAMA-G97 to depalmitoylate α1C-IL (Fig. 3C, D). We conclude from these experiments that chemogenetic control of individual protein palmitoylation using LAMAs is feasible.

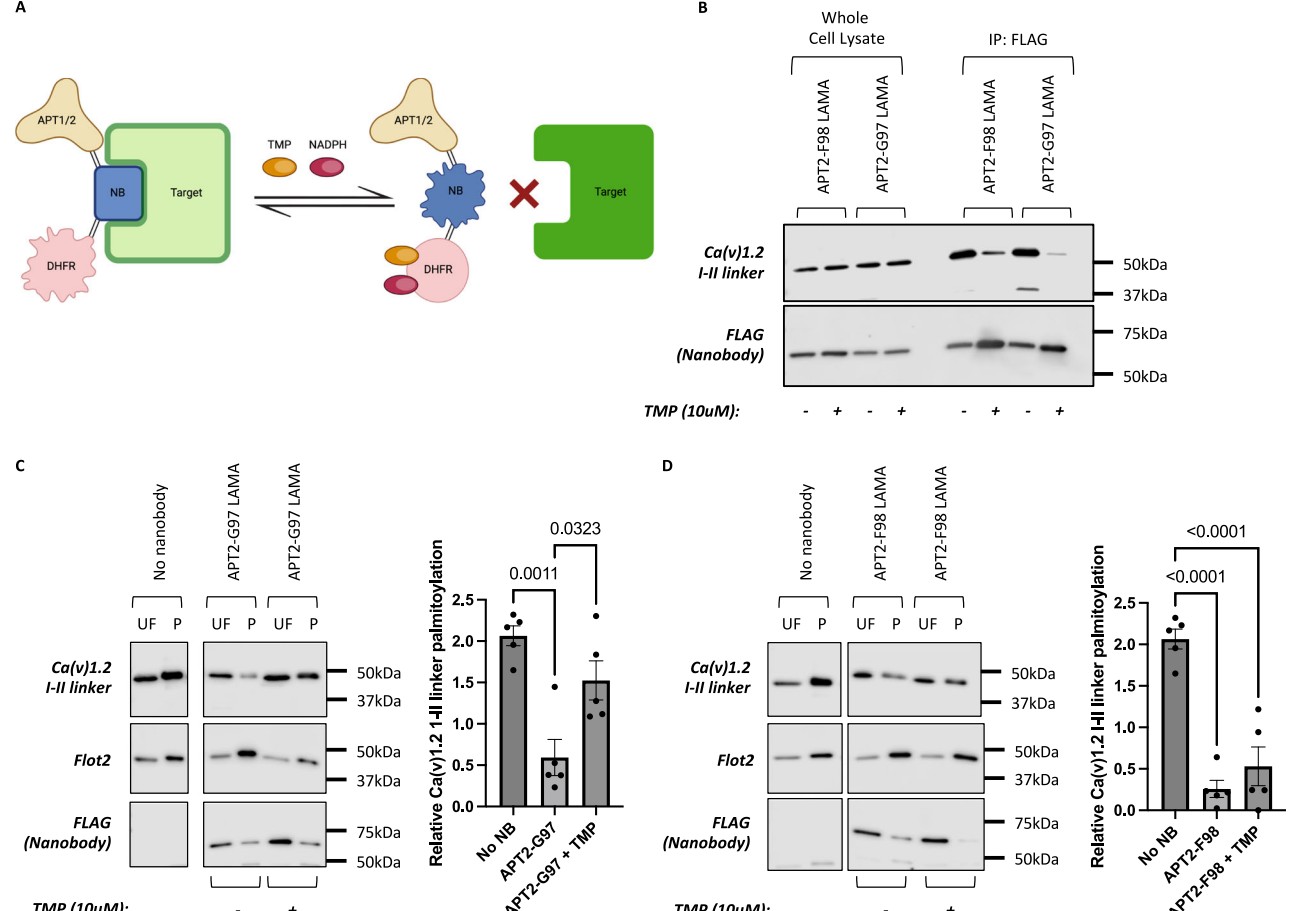

**Fig. 3 | Ligand-modified antibody fragments (LAMAs) to target protein palmitoylation. A** LAMA schematic (Created in BioRender. Fulton, H. (2025) https://BioRender.com/v66u239). **B** Impact of trimethoprim on LAMA binding to YFP-Ca(v)1.2 I-II-linker assessed by immunoprecipitating the FLAG tagged nanobody. This experiment was repeated independently three times with similar results. **C** Inserting dihydrofolate reductase (DHFR) at position G97 in CDR3 of the GFP enhancer nanobody generates a LAMA with potent depalmitoylating activity

towards YFP-Ca(v)1.2 I-II-linker. Treatment with 10 µM trimethoprim restores target protein palmitoylation. Data are means ± SEM from $n = 5$ independent experiments. **D** Inserting DHFR at position F98 in CDR3 of the GFP enhancer nanobody generates a LAMA with potent, trimethoprim-insensitive depalmitoylating activity towards YFP-tagged Ca(v)1.2 I-II-linker. Data are means ± SEM from $n = 5$ independent experiments. Statistical comparisons: one-way ANOVA followed by Tukey's multiple comparisons test. Source data are provided as a Source Data file.

To visualise nanobody chimera-mediated protein redistribution in cells we selected a model substrate lacking integral membrane domains whose cellular location changes from membrane anchored to cytosolic when it is depalmitoylated. Sprouty-2 (Spry2) is an antagonist of receptor tyrosine kinase signalling which is palmitoylated by Golgi-localised zDHHC-PATs[32]. APT2-LAMA-G97 efficiently depalmitoylated eGFP-Spry2 in the absence but not presence of TMP (Fig. 4A). We visualised eGFP-Spry2 cellular localisation and how it was remodelled by APT2-LAMA-G97. When expressed alone, eGFP-Spry2 localised predominantly to punctate intracellular structures, consistent with it being palmitoylated in the Golgi (Fig. 4B). Co-expressing APT2-LAMA-G97 was largely without impact on eGFP-Spry2 cellular distribution if TMP was co-applied overnight (Fig. 4B) but changed it to entirely cytosolic in the absence of TMP (Fig. 4B).

We evaluated how quickly APT2-LAMA-G97 depalmitoylates its target when TMP is withdrawn to allow the nanobody to bind and deliver APT2. We first used a simple fractionation approach to evaluate how quickly eGFP-Spry2 was released from membranes following TMP withdrawal. In cells transfected with APT2-LAMA-G97, a small amount of cytosolic eGFP-Spry2 was detected in the presence of TMP (Fig. 4C). Remarkably, one hour after withdrawing TMP, eGFP-Spry2 was entirely localised to the cytosolic fraction to a level indistinguishable from eGFP-Spry2 co-expressed with APT2-LAMA-G97 in the absence of TMP (Fig. 4C). We therefore used live cell imaging to monitor eGFP-Spry2

redistribution upon TMP withdrawal. The punctate intracellular distribution of eGFP-Spry2 rapidly changed to a diffuse cytosolic pattern 15–45 min after removal of TMP (Fig. 4D and Supplementary Movies 1–3).

Although we detected no change in palmitoylation of the house-keeping protein flotillin-2 when APT2-LAMA-G97 was induced to release its target using TMP (Fig. 3D), we reasoned that TMP-induced release of this agent from its target might promote off-target depalmitoylation by the newly-released thioesterase. To investigate this possibility, we generated Flp-In T-REx cells stably expressing tetracycline inducible APT2-LAMA-G97 and quantified both whole-proteome (using TMT labelling) and palmitoyl-proteome (using label-free proteomics) in the presence and absence of tetracycline.

In the whole-proteome analysis a total of 4155 proteins were initially identified. After applying filters to exclude proteins identified by site only, reverse sequences, potential contaminants, proteins identified by fewer than two peptides, proteins with fewer than two valid values per group, and duplicate unreviewed protein IDs, the final dataset consisted of 2869 proteins. After applying a log2 fold change cut-off of 1.0 and a q-value threshold of 0.05, the only overexpressed protein identified was LYPLA2 (APT2, a component of the nanobody chimera) with a log2 fold change of 1.5 (Supplementary Fig. 2).

We identified 1119 proteins in the palmitoyl proteome, which was reduced to 504 proteins after filtering. Among a total of 27

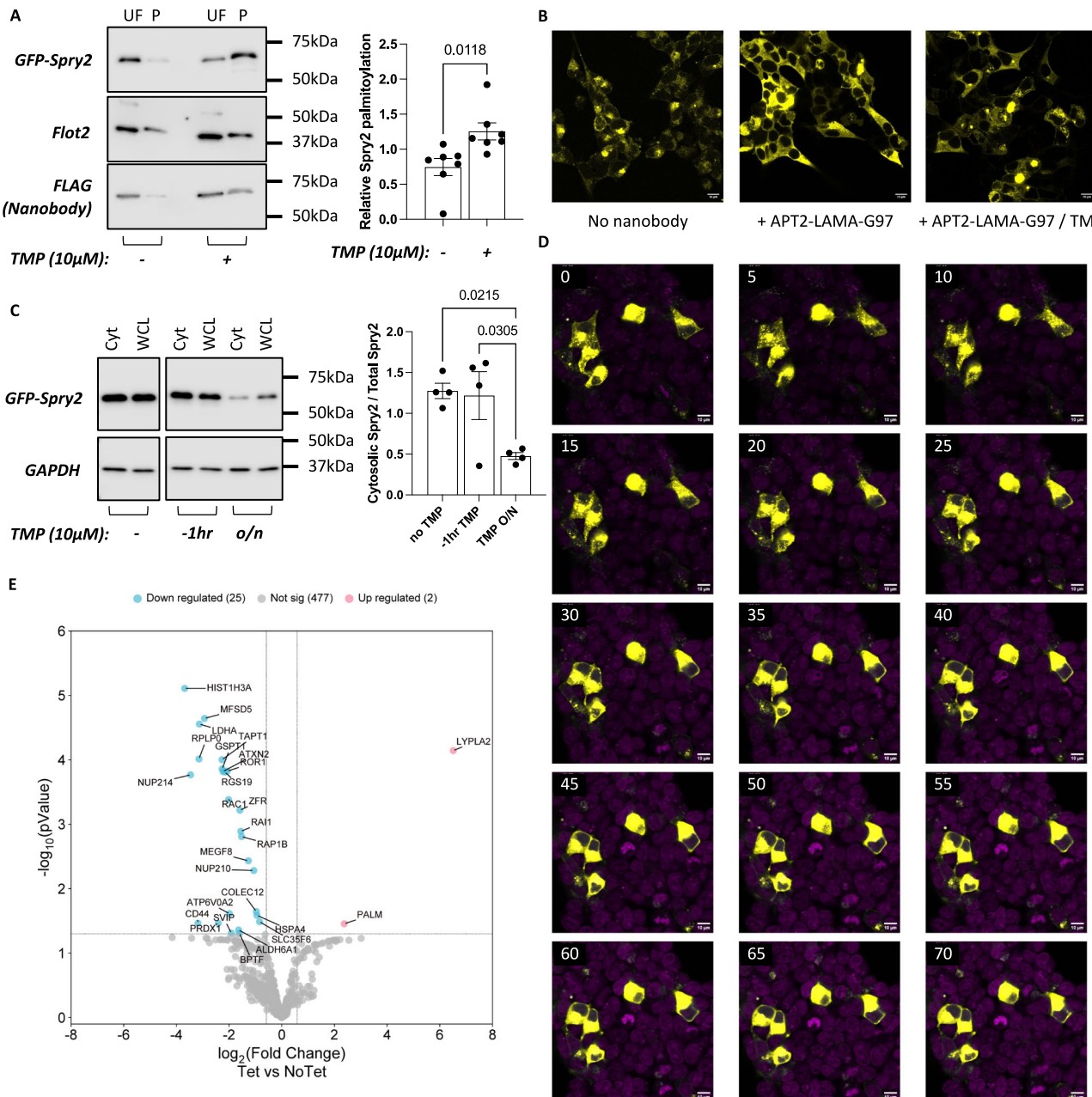

**Fig. 4 | Impact of LAMA-mediated depalmitoylation on cellular distribution of GFP-Spry2 in HEK293T cells. A** Impact of trimethoprim (TMP, 10 µM, applied overnight) on APT2-LAMA-G97 mediated depalmitoylation of eGFP-Spry2, assessed using acyl-RAC. Data are means ± SEM from *n* = 7 independent experiments. Statistical comparison: unpaired two-tailed t-test. **B** Subcellular distribution of eGFP-Spry2 in the absence of nanobody (left), the presence of nanobody (middle), and the presence of nanobody and trimethoprim (10 µM, applied overnight). Scale bar: 10 µm. Experiment was repeated independently five times with similar results. **C** Impact of trimethoprim withdrawal on eGFP-Spry2 cellular distribution assessed by fractionation. Cyt: cytoplasmic fraction, WCL: whole cell lysate, -1hr: trimethoprim (10 µM, applied overnight) then withdrawn for 1 h, o/n: trimethoprim (10 µM) applied overnight. Data are means ± SEM from *n* = 8 independent experiments. Statistical comparisons: one-way ANOVA followed by Dunnett's multiple comparisons test. **D** Live cell imaging to visualise eGFP-Spry2 redistribution upon trimethoprim withdrawal. Snapshots taken at 5 min intervals following withdrawal of trimethoprim from cells expressing eGFP-Spry2 and APT2-LAMA-G97. The number if the upper left corner of each image indicates the time (minutes) since TMP withdrawal. Scale bar: 10 µm. Experiment was repeated independently five times with similar results. **E** Palmitoyl proteome analysis of Flp-In T-REx cells stably expressing tetracycline-inducible APT2-G97-LAMA. Data are presented as log2 fold change values with a cut-off of 0.58 and a q-value threshold of 0.05. Statistical comparisons: moderated, two-sided t-tests with adjustments for multiple comparisons using the Benjamini-Hochberg procedure. Palmitoylation of 25 proteins (5% of the palmitoyl proteome) is reduced and palmitoylation of 2 proteins is increased following induction of the nanobody chimera expression. Data from *n* = 3 independent experiments. Source data are provided as a Source Data file.

differentially palmitoylated proteins following expression of APT2-LAMA-G97, 25 displayed reduced palmitoylation and 2 (including LYPLA2) displayed enhanced palmitoylation (Fig. 4E). We conclude from these experiments that expression of nanobody-thioesterase chimeras does not remodel the cellular proteome, and modestly remodels a small fraction of the cellular palmitoyl proteome.

## Nanobody-mediated depalmitoylation of Ca(v)1.2

Mutating palmitoylation sites in the Ca(v)1.2 α1C subunit induces a -10 mV rightward shift in the half-maximum activation voltage (V50) of Ca(v)1.2[17]. Achieving such a shift in the Ca(v)1.2 V50 is desirable therapeutically because it would suppress arrhythmias by reducing the susceptibility of ventricular myocytes to early after-depolarisations

(EADs) by reducing the Ca(v)1.2 window current[33]. We therefore set out to target Ca(v)1.2 α1C palmitoylation using nanobodies. We fused APT2 to nanobody F3 which binds the Ca(v)1.2 beta subunit[34]. F3 binding to Ca(v)1.2 is functionally silent, but when fused to an E3 ubiquitin ligase this nanobody ubiquitinates Ca(v) alpha subunits in a beta-subunit dependent manner[34].

We co-expressed APT2-F3 with Ca(v)1.2 beta subunits in 293 T-REx cells stably expressing Ca(v)1.2 α1C subunit. Immunoprecipitating APT2-F3 co-purified Ca(v)1.2 α1C subunit only when either β2ba or β2b subunits were co-expressed (Fig. 5A). APT2-F3 depalmitoylated Ca(v)1.2 α1C subunit only when co-transfected with a beta subunit (Fig. 5B). We conclude from these experiments that a nanobody directed towards the beta subunit can deliver a depalmitoylating enzyme to the Ca(v)1.2 alpha subunit.

We evaluated nanobody impact on Ca(v)1.2 function by voltage clamping cells engineered to express wild type α1C that were transfected with β2b and α2δ1 subunits, plus either APT2-F3 or catalytically inactive (S122A) APT2-F3. Transfecting cells expressing α1C with APT2-F3 shifted the voltage dependence of activation of Ca(v)1.2 rightwards by ~10 mV compared to cells transfected with catalytically inactive APT2-F3 (Fig. 5C–F). This mirrors the functional effect observed using mutagenesis to prevent palmitoylation of the Ca(v)1.2 α1C subunit[17]. We conclude that the voltage dependence of activation of Ca(v)1.2 can be specifically tuned using a Ca(v)1.2-binding nanobody fused to a thioesterase.

In monolayers of human induced pluripotent stem cell derived cardiomyocytes (iPSC-CMs), blocking the hERG channel reduces the magnitude of the repolarising current IKr and increases the action potential duration (APD). This prolongs the plateau phase of the action potential, which increases depolarising Ca(v)1.2 window currents, further extending APD and generating EADs (which trigger the lethal arrhythmias ventricular tachycardia and ventricular fibrillation in vivo). These features are the basis for the use of iPSC-CMs to detect pro-arrhythmic actions of drugs[35]. Interventions that reduce Ca(v)1.2 window current magnitude (for example depalmitoylating Ca(v)1.2 α1C subunit) will blunt the APD prolonging effects of HERG blockade. We expressed APT2-F3 or catalytically inactive APT2-F3 in iPSC-CMs, recorded action potentials using the voltage sensitive dye fluovolt, and treated cells with the HERG channel blocker dofetilide (10 nM) to prolong APD and evaluate EAD susceptibility. Action potentials were indistinguishable in iPSC-CMs expressing APT2-F3 or catalytically inactive APT2-F3 before treating with dofetilide (Fig. 6A, B). Expressing APT2-F3 did not adversely impact iPSC monolayer viability compared to catalytically inactive APT2-F3 (Supplementary Fig. 5). Dofetilide-induced APD prolongation was significantly blunted in cells expressing APT2-F3 compared to catalytically inactive APT2-F3 (Fig. 6A–C). In addition, dofetilide-induced EAD frequency was significantly reduced in iPSC-CM monolayers expressing APT2-F3 (5/21 EAD-free) compared to catalytically inactive APT2-F3 (0/21 EAD-free, Fig. 6D). These experiments demonstrate the feasibility of targeting Ca(v)1.2 α1C subunit palmitoylation to suppress lethal cardiac arrhythmias and highlight the potential therapeutic applications of enzyme-nanobody chimeras.

## Discussion

We describe a panel of genetic reagents that specifically depalmitoylate individual proteins using a targeting module based on a nanobody. This approach successfully targets palmitoylation of proteins using nanobodies with sub-nanomolar (0.7 nM, LaG-16[15]), single-digit nanomolar (G97 and F98 LAMA[30]), and low nanomolar (13 nM, F3[34]) affinities for their targets. Nanobody-mediated depalmitoylation is efficacious using nanobodies that bind directly to a protein target and also to target palmitoylation of proteins in a multi-protein complex that interact with the nanobody's target, which suggests that the effective radius of depalmitoylation of these reagents is relatively large.

Nanobodies are powerful tools with numerous applications in biology and medicine. Nanobodies fused to E3 ubiquitin ligases induce ubiquitination and degradation of their targets and/or associated proteins[34,36,37], while nanobodies fused to deubiquitinases deubiquitinate and stabilise their targets[38]. Nanobodies conjugated to enzymes have also been employed to manipulate protein O-GlcNAcylation[39,40] and phosphorylation[41]. Here we expand the repertoire of PTMs that can be targeted using this technology. Considering Ca(v)1.2, which has previously been successfully inhibited using nanobodies fused to E3 ubiquitin ligases[34,36], our results confirm the feasibility of tuning the biophysical properties of this channel and consequently reducing arrhythmia susceptibility by targeting its palmitoylation in cardiac myocytes. Enzyme-conjugated nanobodies capable of shifting the properties of voltage sensitive channels such as Ca(v)1.2 or Na(v)1.5[42] consequently offer promise treating cardiac arrhythmias as well as being powerful and specific experimental tools.

Chemical inducers of proximity (CIPs) allow precise and temporal control of numerous biological processes[43]. Strategies to induce proximity between proteins using small molecules most commonly employ rapamycin based targeting (for example[44–47]). The specificity of this approach is excellent, but the high affinity of rapamycin and its analogues essentially renders the proximity that they induce irreversible[48]. Alternative, lower affinity and reversible CIPs have been described[49,50]. The TMP-sensitivity of our APT2-LAMA chimera clearly offers advantages over approaches that simply conjugate enzymes to nanobodies, and we suggest it is also superior to rapamycin-based CIP approaches because it enables rapid, dynamic, and reversible control of target protein palmitoylation. This system is similar in performance to a related nanobody-based induced proximity system, which is also rapidly reversed by exogenously applied TMP[51].

Our success using nanobodies to induce protein depalmitoylation implies that bifunctional molecules capable of recruiting a thioesterase to a target protein would induce that protein's depalmitoylation, in the same manner that PROTACs induce protein degradation by recruiting E3 ubiquitin ligases[52], and PHOSTACs induce dephosphorylation by recruiting phosphatases[53]. This will be the subject of future investigations. The principal limitation of the nanobody-based approach is the off-target depalmitoylation of a limited number of proteins by our APT2-LAMA chimera. It is conceivable that this off-target activity may adversely affect cell viability. However, since the only protein to change abundance in cells stably expressing nanobodies was the chimera itself (and cellular stress typically manifests as dynamic and co-ordinated remodelling of expression of several hundred proteins) we suggest it is unlikely that the modest off-target activity of our chimeras impairs cell viability. We anticipate that by engineering TMP sensitivity into the nanobody's cargo (the thioesterase) as well as the nanobody itself that future experiments will improve the specificity of these reagents by inhibiting the enzymatic activity of nanobody chimeras released from their targets by TMP.

Nanobody-mediated depalmitoylation successfully targets proteins that are not usually substrates of the thioesterase fused to the nanobody (e.g. APT2 targeting Ca(v)1.2 and PLM) and enhances the ability of a thioesterase to depalmitoylate an established substrate (e.g. APT1 and NCX1). This implies that in a cellular setting the ability of a thioesterase to depalmitoylate a substrate is largely governed by as-yet undetermined selectivity 'rules' of that enzyme for its substrate, rather than an inherent ability of one particular enzyme to hydrolyse a thioester bond on one protein but not another. The rules determining substrate recognition by palmitoylating enzymes are emerging[13,14,54], and our understanding of these rules will be improved by chemical-genetics approaches to identify more substrates[55,56]. Our results imply that rules governing thioesterase recognition of substrates also exist but remain essentially unknown. Nonetheless, we acknowledge that some selectivity exists in the tractability of nanobody mediated depalmitoylation. For example, even when fused to a nanobody, APT1

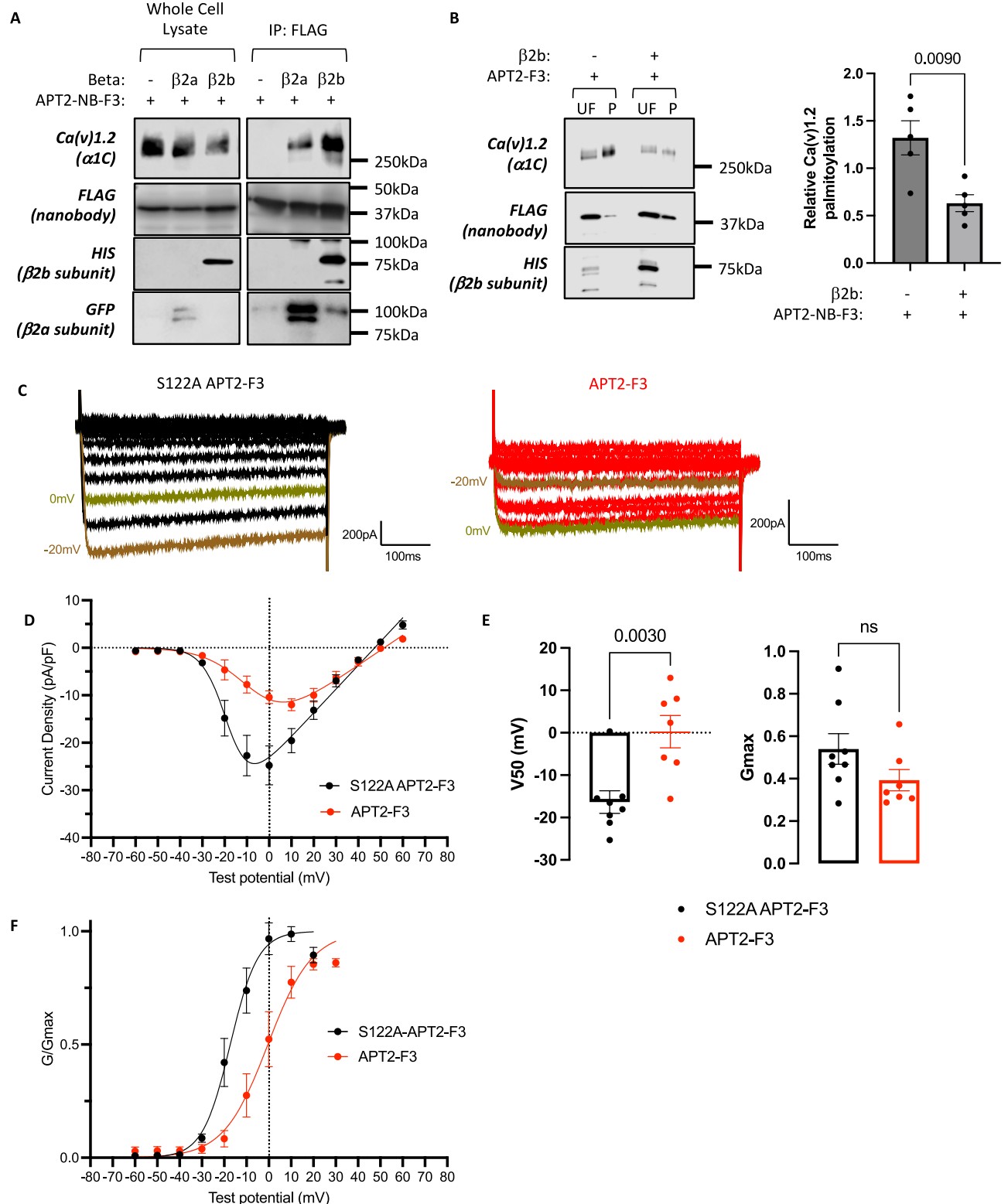

cannot depalmitoylate PLM, and APT2 cannot depalmitoylate NCX1. We do not rule out steric factors such as linker length between nanobody and enzyme as contributing to this selectivity. This will be the subject of future investigations.

In a cell-free setting, saturating concentrations of NADPH and trimethoprim entirely abolish binding of both G97 and F98 LAMA to GFP[30]. In our experiments using intact cells, we detected residual binding of F98 LAMA to YFP in the presence of 10 μM trimethoprim.

This manifested functionally as continued depalmitoylation of the YFP fusion protein, which we interpret to mean that the binding affinity of F98 LAMA for its target was not sufficiently reduced by trimethoprim. In contrast, trimethoprim-induced release of APT2-G97-LAMA from YFP wholly restored palmitoylation of its target protein to a level not distinguishable from cells not expressing nanobody. The affinity range across which nanobodies effectively deliver depalmitoylating activity to a target protein remains to be established but will be an important

**Fig. 5 | Nanobody-mediated depalmitoylation of Ca(v)1.2 α1C subunit in engineered cell lines. A** APT2-Nanobody-F3 fusion protein recruitment to the Ca(v)1.2 complex assessed by immunoprecipitating the FLAG tagged nanobody. Experiment was repeated independently three times with similar results. **B** APT2 fused to Ca(v)1.2 beta subunit nanobody F3 depalmitoylates Ca(v)1.2 α1C subunit only when co-expressed with HIS-tagged beta2b subunit to facilitate nanobody recruitment to Ca(v)1.2. Data are means ± SEM from $n = 5$ independent experiments. Statistical comparison: unpaired two-tailed t-test. **C** 293 T-REx cells stably expressing α1C and transfected with β2b and α2δ1 subunits, plus either APT2-F3 or catalytically inactive (S122A) APT2-F3, were voltage-clamped at −60 mV. 500 ms voltage steps from −60 mV to +60 mV at 10 mV increments were applied at 1 Hz. Representative examples of currents from cells expressing inactive (black, left) or active (red, right) APT2 fused to nanobody F3 with barium as charge carrier are shown. **D** Current

voltage relationship for Ca(v)1.2 mediated barium currents in the presence of active APT2-F3 (red, data are means ± SEM from $n = 9$ independent experiments) or catalytically inactive APT2-F3 (black, data are means ± SEM from $n = 8$ independent experiments). **E** A Boltzmann function was fitted to the I−V relationships to yield V50 and Gmax parameters for catalytically inactive APT2-F3 (black, data are means ± SEM from $n = 8$ independent experiments) or APT2-F3 (red, data are means ± SEM from $n = 7$ independent experiments). Statistical comparison: unpaired two-tailed t-test. **F** G−V curves were drawn by calculating G/Gmax using parameters obtained from the Boltzmann fits of I−V curves from individual cells. Data are means ± SEM from $n = 8$ (catalytically inactive APT2-F3, black) and $n = 7$ (APT2-F3, red) independent experiments. Statistical comparisons: unpaired t-test. Source data are provided as a Source Data file.

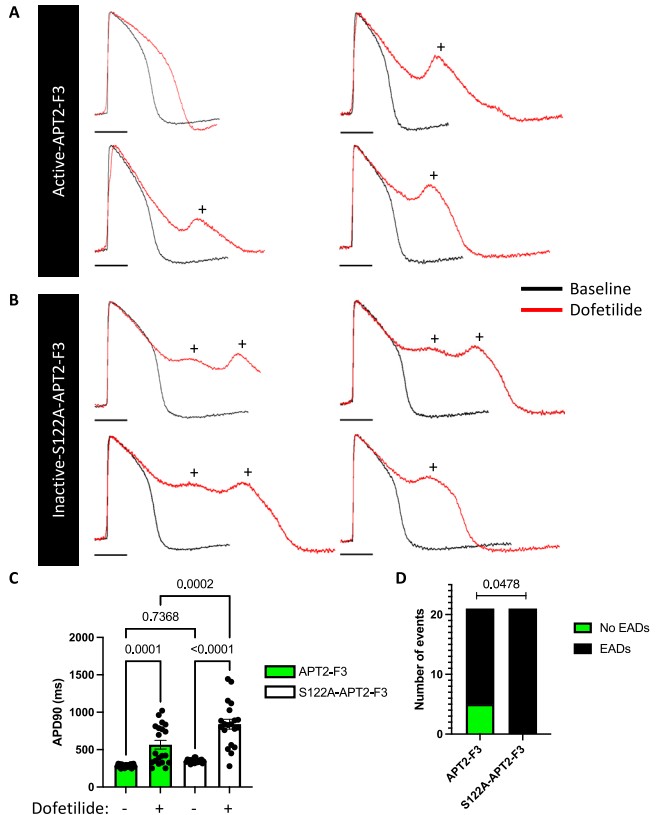

**Fig. 6 | Nanobody-mediated depalmitoylation of Ca(v)1.2 α1C subunit in induced pluripotent stem cell derived cardiac myocytes.** Representative action potential recordings from induced pluripotent stem cell derived cardiac myocytes (iPSC-CMs) expressing APT2-Nanobody-F3 **A** or catalytically inactive (S122A) APT2-Nanobody-F3 **B** before (black traces) and 30 min after (red traces) the addition of dofetilide (10 nM) to prolong action potential duration. Early after depolarisation-like events (EADs, +) were detected using an automated algorithm. Scale bar: 200 ms **C** Action potential durations in iPSC-CMs expressing APT2-F3 or catalytically inactive (S122A) APT2-F3 before and after treatment with dofetilide (10 nM, 30 min). Action potential durations (APDs) are indistinguishable in iPSC-CMs transfected with active or inactive APT2 fused to F3. Application of dofetilide prolongs APD to a significantly greater extent in cells expressing S122A-APT2-F3 compared to cells expressing APT2-F3. Data are means ± SEM from $n = 20$ independent experiments. Statistical comparisons: one-way ANOVA followed by Tukey's multiple comparisons test. **D** Dofetilide-induced EAD-like event incidence in iPSC-CM monolayers expressing APT2-F3 compared to catalytically inactive S122A APT2-F3. Statistical comparison: Two-sided Fisher's Exact Test ($N = 21$). Source data are provided as a Source Data file.

future question to address to develop these reagents further. Nanobodies binding many different classes of proteins have been described and catalogued[57], and new nanobodies are readily identified using library-based approaches[58].

In conclusion, we describe and validate an approach to specifically tune the function of palmitoylated proteins. Nanobody-thioesterase chimeras show promise as experimental tools to enable dynamic manipulation of protein palmitoylation and lay the foundations for clinical applications for specific depalmitoylating agents to ameliorate cardiac arrhythmias.

## Methods
### Ethical statement
This study utilised primary ventricular myocytes from rabbits. All protocols involving animals were approved by the University of Glasgow Animal Welfare and Ethics Review Board. Rabbit hearts were excised from terminally anaesthetized, heparin-treated animals under the authority of Project Licence PP5254544 granted by the UK Home Office.

### Plasmids and molecular biology
pET21-pelB-LaG-16 was a gift from Michael Rout (Addgene plasmid #172746, RRID:Addgene_172746)[15]. pET51b(+)_LAMA-F98 and pET51b(+)_LAMA-G97 were a gift from Kai Johnsson (Addgene plasmids #130714 and #130715, RRID:Addgene_130714 and RRID:Addgene_130715)[30]. Plasmids encoding anti-Ca(v)1.2 nanobody F3, and Ca(v)1.2 α2δ1 and β2b subunits were kindly provided by Professor Henry Colecraft, Columbia University, USA. A plasmid encoding eGFP-Sprouty-2 was generously provided by Professor Luke Chamberlain, University of Strathclyde, UK. All new plasmids and point mutations were generated using InFusion cloning (Takara) employing primers designed using the Takara online primer design tool.

### Antibodies and western blotting
This investigation used an anti-Ca(v)1.2 α1C subunit antibody raised in guinea pig obtained from Alomone Labs, and antibodies against flotillin 2 from BD Biosciences, GAPDH, FLAG and HIS tags from Merck, and GFP from Protein Tech. Images of western blots were acquired and analysed using Image Studio Version 5.2 and an Odyssey-FC Imager (LI-COR Biosciences).

### Cell culture
Calcium-tolerant, adult rabbit ventricular myocytes were isolated from the left ventricle of male New Zealand White rabbits (2.8–3.2 kg, Envigo), following perfusion with collagenase in the Langendorff mode. Ventricular myocytes were plated onto laminin-coated 35 mm dishes for 2 h before experimentation.

HEK293T cells (ATCC) and Flp-In 293 T-REx cells (Thermo Fisher Scientific) stably expressing tetracycline-inducible Ca(v)1.2 α1C subunit were cultured and transfected as previously described[17]. A Flp-In 293 T-Rex cell line stably expressing tetracycline-inducible APT2-LAMA-G97 was generated as previously described[17]. Briefly, APT2-LAMA-G97 was subcloned into pcDNA5-FRT/TO and cotransfected with pOG44 Flp-Recombinase (ThermoFisher Scientific) into Flp-In 293

T-REx cells. Stably transfected cells were selected and maintained using hygromycin and blasticidin (Invivogen) according to the manufacturer's instructions.

Cells were transfected using Lipofectamine 2000 (ThermoFisher Scientific) in 6-well and 12-well culture plates and harvested 18–24 h after transfection. When cells were treated with trimethoprim, the transfection mix was removed and replaced with culture media containing trimethoprim (Merck, 10 μM) or vehicle (0.1% v/v DMSO) 4 h after transfection.

### Assessment of protein palmitoylation by resin-assisted capture

Palmitoylated proteins were purified from whole cell lysates 24 h after transfection using acyl-resin assisted capture (Acyl-RAC). Free thiols were first alkylated with methyl methanethiosulfonate (1% v/v) in 2.5% SDS, 100 mM HEPES, 1 mM EDTA pH 7.4. After acetone precipitation and resolubilising in 1% SDS, 100 mM HEPES, 1 mM EDTA pH 7.4, palmitoylated proteins were captured using thiopropyl-Sepharose in the presence of 250 mM neutral hydroxylamine[59].

### Co-immunoprecipitation

Cells were lysed in in PBS supplemented with 1% (v/v) Triton X-100, 0.1% (w/v) SDS, and a protease inhibitor cocktail (Merck). Insoluble material was removed by centrifuging at 17,000 g for 5 min at 4 °C. FLAG tagged proteins were captured with anti-FLAG affinity resin (Merck). Beads were washed five times with 1% Triton X-100 in PBS and proteins eluted using SDS-PAGE loading buffer.

### Cell fractionation

Cells were solubilised in 0.05% w/v digitonin in PBS supplemented with protease inhibitors for 30 min at 4 °C. Lysates were centrifuged at 17,000 g for 5 min at 4 °C and the supernatant collected and designated 'cytosolic fraction'.

### Microscopy

HEK293T cells cultured on glass coverslips were co-transfected with eGFP-SPRY2 and APT2-LAMA-G97. Four hours after transfecting, cells were treated with 0.1% of DMSO (vehicle) or 10 μM TMP overnight. The next day, cells were washed with PBS, then fixed and mounted on glass slides in mounting media supplemented with DAPI. The images were acquired using a Zeiss LSM880 confocal microscope with x63 oil-immersion objective. Excitation and emission were set to 488 nm and 530 nm respectively for GFP and 405 nm and 449 nm for DAPI.

For live cell imaging, TMP (10 μM) was added to cells 4 h after the cells were co-transfected with GFP-SPRY2 and APT2-LAMA-G97. The next day, two drops of NucRed Live 647 ReadyProbes Reagent (Thermo Fisher Scientific) were added 30 min before acquiring images of live cells. Culture media containing TMP was removed from the cells and replaced with phenol red free media, and recordings acquired immediately. Images were recorded using a Nikon inverted AX-R microscope using x20 objective at 37 °C and 5% CO$_2$ with GFP channel (488/491–550 nm) and far-red channel (640/647-700 nm).

### Protein digestion and peptide extraction

Global proteome after expression of APT2-LAMA-G97: A 100 μg protein sample was dissolved in 4 M urea in 50 mM triethylammonium bicarbonate (TEAB) and 200 mM tris(2-carboxyethyl)phosphine (TCEP) buffer. The proteins were denatured by incubation at 55 °C for 1 h, followed by alkylation with 375 mM iodoacetamide. Tryptic digestion was carried out using Trypsin Gold (Promega, UK) at an enzyme-to-protein ratio of 1:25, with the reaction incubated overnight at 37 °C. Tryptic digested peptides were labelled with TMT sixplex reagent (n = 3 per group) according to the manufacturer's instructions (Thermo Fisher, UK). The labelled proteins were pooled and vacuum-dried, then subjected to fractionation using a high pH reversed-phase fractionation kit (Thermo Fisher, UK). Nine fractions were collected, vacuum-dried, and each fraction was analysed separately by LC-MS/MS, following the procedure outlined below.

Palmitoylated proteome after expression of APT2-LAMA-G97: Palmitoylated proteins purified using acyl-RAC (n = 3 per group) were eluted from Sepharose beads using SDS-PAGE loading buffer. The samples were briefly separated on a 10% SDS-PAGE gel for 5 min at 125 V in MOPS buffer. Gel bands measuring ~50 mm² were excised and cut into 1 mm² pieces. The bands were washed with 100 mM ammonium bicarbonate (AmBic) followed by a wash with 50% acetonitrile (ACN)/AmBic. Proteins were reduced with dithiothreitol (DTT), alkylated with iodoacetamide (IAA), and digested with trypsin. Peptides were extracted using 50% ACN/0.2% formic acid and vacuum-dried.

### Liquid chromatography-tandem mass spectrometry

LC-MS/MS was conducted using an UltiMate 3000 nano-flow system (Dionex/LC Packings, USA) coupled to an LTQ Orbitrap Velos FTMS hybrid mass spectrometer (Thermo Fisher Scientific, Germany) equipped with a nano-electrospray ion source. A 5 μL sample was first loaded onto a Dionex C18 nano trap column (0.1 × 20 mm, 5 μm) at a flow rate of 5 μL/min in a solvent mixture of 98% water, 0.1% formic acid, and 2% acetonitrile. The sample was subsequently eluted onto an Acclaim PepMap C18 nano column (75 μm × 50 cm, 3 μm, 100 Å) at a flow rate of 0.3 μL/min. Both the trap column and the nano-flow column were maintained at 35 °C. The elution was performed with a linear gradient of solvent A (98:2 water/acetonitrile with 0.1% formic acid) against solvent B (20:80 water/acetonitrile with 0.1% formic acid). Ionisation was carried out in positive ion mode using a Proxeon nano-electrospray ESI source (Thermo Fisher, UK), and the ions were analysed using the Orbitrap Velos FTMS (Thermo Fisher, Germany). The column was washed and re-equilibrated between injections. The ionisation voltage was set to 2.8 kV, and the capillary temperature was 250 °C. The mass spectrometer operated in data-dependent MS/MS mode, scanning a mass range of 380–1600 amu. The 20 most abundant multiply charged ions from each full scan were selected for MS/MS analysis using high-energy collision dissociation (HCD) at 35% collision energy. The mass resolution was set to 60,000 for MS1 and 7500 for HCD MS2. MS raw data files were searched against the Uniprot Human database (UP000005640, downloaded 25/04/2024) using MaxQuant v2.5.2.0 algorithm[60]. Further data filtering was performed using Perseus version 2.1.1.0[61] and LFQAnalyst[62] for label-free quantification, while TMT-labelled data analysis was conducted using Provision software[63]. Statistical analysis was performed using the limma package (R Bioconductor within the software), which applies moderated t-tests for pairwise comparisons. Tests were two-sided, and adjustments for multiple comparisons were made using the Benjamini-Hochberg procedure. Quality metrics of the whole proteome and palmitoyl proteome analyses are presented in Supplementary Figs. 3 and 4. All protein identifications with abundances are provided in Supplementary Data 1.

### Whole cell voltage clamping

Whole cell patch clamp was used to record Ca(v)1.2 mediated Ba$^{2+}$ currents at room temperature from cells stably expressing tetracycline inducible α1C, 24–48 h after co-transfecting with β2b (2 μg), α2δ1 (0.4 μg), nanobody (1.6 μg) and CD8 (0.25 μg) using lipofectamine 2000 (6 μl) in a 35 mm dish. Dynabeads coupled with an anti-human CD8 antibody (Invitrogen) were used to visually identify cells which were successfully transfected.

Membrane currents were recorded using Clampex 10.3 (Molecular Devices) and analysed using Clampfit 10.7 (Molecular Devices) as described previously[17]. Briefly, recordings were conducted from a holding potential of −60 mV using 500 ms voltage pulses at 1 Hz. The pipette solution contained 120 mM NMDG-Cl, 1 mM MgCl$_2$, 5 mM EGTA, 4 mM Mg-ATP, 42 mM HEPES (pH 7.3 adjusted with

methanesulfonic acid). Bath solution contained 40 mM $BaCl_2$, 1 mM $MgCl_2$, 105 mM Tris (pH 7.3 adjusted with methanesulfonic acid).

I-V curves were fitted to the Boltzmann Eq. (1) using GraphPad Prism 10.1.1:

$$I = Gmax(Vm - Vrev)/(1 + \exp(-(Vm - V50)/Ka)) \quad (1)$$

Gmax is the maximal conductance, Vm is the membrane voltage, Vrev is the reversal potential of the current, Ka is the slope factor, and V50 is the half-maximum activation voltage.

The values obtained for Gmax and Vrev were used to calculate fractional conductance values for each cell at each Vm using Eq. (2):

$$G/Gmax = I/(Gmax(Vm - Vrev)). \quad (2)$$

Conductance-voltage (G-V) curves were fitted to the form of the Boltzmann Eq. (3):

$$G/Gmax = 1/(1 + \exp(-(Vm - V50)/Ka)). \quad (3)$$

### Action potential recordings from induced pluripotent stem cell derived cardiac myocytes

ICell2 (Cellular Dynamics International, Cat# R1218, purity >95% cardiomyocytes) were seeded on glass, coated with fibronectin (10 μg/ml, bovine, Gibco, Cat# 33010018) in a 96-well plate. The seeding density advised by the manufacturers was $1.5 \times 10^5$ cells/cm² for ICell2.

Before experimentation hiPSC-CM monolayers were inspected and scored to estimate their viability (monolayer scores: 0: intact monolayer, 1: some small holes appearing, 2: larger and more frequent holes, 3: peeling/cell death). HiPSC-CMs were incubated in serum-free medium (Fluorbrite™ DMEM, ThermoFisher) for at least 1 h and subsequently loaded with FluoVolt Dye (1:1000, Invitrogen, Cat# F10488) and Powerload Concentrate (1:100, Invitrogen, Cat# F10488) for 20 min at 37 °C. FluoVolt fluorescence was recorded on the CellOPTIQ system[64] using a 40× objective, 470 nm LED, photomultiplier tube (PMT)/amplifier system and the signal was acquired at 10 kHz and stored for subsequent analysis. 96 well plates were maintained at 5% $CO_2$ and 37 °C using an on-stage incubator. The fluorescence signals representing membrane voltage were analysed using CellOPTIQ software, which provided an averaged action potential (AP) waveform from a minimum of 3 sequential AP complexes and calculated the set of parameters related to the AP time course.

### Statistics and reproducibility

Sample sizes were chosen based on power calculations (80% power, 25% difference between groups, co-efficient of variation 0.1). No data was excluded from the analysis. All data presented has utilised numerous biological replicates. Cultured cells were randomised to treatment groups in multi-well dishes at the time of plating. Investigators were not blinded during data collection or analysis.

All data are presented as mean ± standard error of the mean. Quantitative differences between groups within an experiment were assessed using two tailed unpaired t-tests (when group sizes were 2) or One-way ANOVA (analysis of variance) followed by appropriate post-hoc tests using GraphPad Prism.

### Reporting summary

Further information on research design is available in the Nature Portfolio Reporting Summary linked to this article.

## Data availability

The MS proteomics data have been deposited to the ProteomeXchange Consortium via the MassIVE partner repository under dataset identifier PXD057626. The UniProt human proteome databases are available online [https://www.uniprot.org/]. Accession codes used in this study: 6LR7. All remaining data generated in this study are provided in the Source Data file. Source data are provided with this paper.

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

## Acknowledgements
We thank Professor Henry M Colecraft PhD (Columbia University) for providing the Ca(v)1.2 beta subunit nanobody F3. We acknowledge support from the British Heart Foundation grant reference numbers PG/19/5/34150 (to W.F.), PG/22/10847 (to W.F., G.L.S., and C.G.), PG/23/11290 (to W.F.).

## Author contributions
Conceptualisation: G.L.S., W.F. Data curation: S.M. Formal Analysis: C.W.K., C.G., H.F., and S.M. Funding acquisition: W.F., G.L.C., and C.G. Investigation: C.W.K., C.G., H.F., E.D.M., S.A., E.K.G., S.M., and W.M. Project administration: W.F. Resources: C.W.K., H.F., A.D.R., F.J., and E.D. Supervision: W.F. Visualisation: H.F. Writing – original draft: W.F. Writing – review and editing: C.W.K., C.G., S.M., W.M., and G.L.S.

## Competing interests
The authors declare no competing interests.
