## [Transparent Peer Review file · Nature Communications]

Nanobody-thioesterase chimeras to specifically target protein palmitoylation

Corresponding Author: Professor William Fuller

Version 0:

Reviewer comments:

Reviewer #1

(Remarks to the Author)

The addition of palmitate to proteins is a frequent post-translational modification that affects as much as 10% of the human proteome, often with important functional consequences. However, the field lacks specific reagents to test its relevance for a given substrate, particularly in a reversible manner. In this manuscript, the authors present a strategy for selectively removing post-translational palmitate modifications from proteins, specifically the alpha subunit of the Ca(v)1.2 channel. Because palmitoylation modifies channel conductance which is dysregulated in disease, new techniques for its removal could have therapeutic implications.

The authors used the nanobody-based recruitment of a cellular depalmitoylation enzyme (APT1 or APT2) to selectively target this channel. Different strategies were first developed as a proof-of-concept. APT2 (or the related protein APT1) was coupled to a GFP-targeted nanobody, and was shown to efficiently depalmitoylate a GFP-tagged, dually-palmitoylated fragment of Cav1.2. As expected, this depalmitoylation required APT catalytic activity, and binding of the nanobody to GTP.

The ability of these nanobody fusions to target other GFP-tagged palmitoylated proteins was also tested. This uncovered substantial differences in enzyme specificity. APT1, but not APT2, was effective on the cardiac sodium/calcium exchanger NCX1. In contrast, APT2 worked well on the Cav1.2 fragment, and had a more modest effect on paralemmin. The reason for this specificity is unclear, and while it could be due to subtle conformational differences that affect substrate access, it could also suggest an inherent selectivity that would be interesting to follow up in future studies.

The authors further used the ligand-modulated antibody fragment (LAMA) approach to make a chemically regulated version of their GFP-directed depalmitoylation construct. Adding a chemical to disrupt nanobody binding led to a substantial recovery of palmitoylation. This is a very useful tool, and I would be very interested to see it used to address a biological question.

Finally, to target Ca(v)1.2 in cells, they fused APT2 to a nanobody that binds the Ca(v)1.2 beta subunit. When Ca(v)1.2 alpha and beta subunits were co-expressed in HEK cells, nanobody-based targeting of APT2 to the Ca(v)1.2 beta subunit resulted in depalmitoylation of the Ca(v)1.2 alpha subunit. Importantly, voltage clamp experiments showed this resulted in a shift in the voltage dependence of activation of Ca(v)1.2 similar to that previously seen when the palmitoylation site was mutated.

Overall, I found the nanobody-targeting approach for selective protein depalmitoylation to be novel and exciting: it has the potential to address important questions about the role of palmitoylation in the cellular environment. I was however disappointed that these tools were not used to gain new biological insights in the current work. Nevertheless, the demonstration that nanobody-targeted depalmitoylases can modulate the activation of Ca(v)1.2 is significant for its potential to spur the development of new therapeutic approaches.

Specific comments:

1. Few details were provided of experiments using the chemically-controlled (LAMA-modified) nanobody. How long were cells treated with trimethoprim, and what were the kinetics of re-palmitoylation?

2. With the exception of the voltage clamp experiments, all experiments used acyl-RAC. This is an appropriate technique to study palmitoylation, and these studies were well-conducted and appropriately interpreted. However, acyl-RAC reports on all sites, even those not normally subjected to dynamic turnover. It was not clear to me if the nanobody constructs only act on the subset of dynamic sites, or also targets static modifications. This could be addressed using other approaches (ie. metabolic labeling with click-able analogs) and additional substrates. The paper could also benefit from use of additional complementary techniques, such as the use of imaging to show if the nanobody caused localization changes for any of the substrates studied.

Reviewer #2

(Remarks to the Author)

Kuo et al. describe the development and functional characterization of intracellularly expressed nanobodies fused to protein depalmitoylases, which they show can depalmitoylate specific transiently co-expressed or stably expressed substrates in cell lines. In the latter case, involving a specific calcium channel subunit, the authors report a functional change in voltage-dependence of the channel. The authors also demonstrate that such depalmitoylation can be temporally controlled by generating a trimethoprim-sensitive version of the nanobody-depalmitoylase fusion protein.

This is an interesting manuscript that has considerable potential relevance not just for the field of protein lipidation, but also for other areas of post-translational regulation. However, the current findings are slightly limited, and the impact of the study would be increased by demonstrating the use of this approach in a more physiological context and/or by providing more evidence of its selectivity and/or advantages compared to similar methods. It is not essential that the authors address all of the issues below (and this reviewer hesitates to request numerous additional experiments), but a more in-depth characterization would make the potential benefits of this system easier to appreciate, particularly for the broad audience that could potentially use this methodology in the future. In particular:

- i) It would be helpful if the authors were to experimentally benchmark their system against other, similar tunable approaches (most obviously rapalog-based dimerization systems). One obvious advantage is that the GFP tag can potentially report a change in localization of the target protein after depalmitoylation in fixed and even live cells. Such a demonstration, particularly for the trimethoprim-inducible version of the nanobody (and perhaps more readily shown for a cytosolic palmitoyl-protein rather than a channel) would greatly strengthen the manuscript and would provide evidence that the method can provide insights that rapalog-based systems cannot.
- ii) Conversely, one of the main advantages of this system (which is also potentially possible with rapalog-binding fusion protein systems) is that a protein of interest can be palmitoylated in cells and then the functional impact of its depalmitoylation can be assessed, potentially on-demand. In contrast, most prior methods rely on a mutant (usually C->A or C->S) that is never palmitoylated in the first place. Surprisingly, the authors do not emphasize this point but may wish to make it more forcefully in the Introduction and/or Discussion
- iii) With (ii) in mind, though, the authors should acknowledge (and cite accordingly) that a similar system was recently developed for protein phosphatases (PMID: 36720221). The current study provides support for the broader applicability of this method, but the statement in the Discussion about the approach being limited to ubiquitination should be modified. As with (i), if the authors have introduced specific enhancements beyond those reported for phosphatase- and/or E3 ligase-based systems these could be highlighted.
- iv) It is unclear if the nanobody itself becomes palmitoylated (apparently yes, in Fig 4B) and how this might affect its efficacy and selectivity. An experimental demonstration of this point (by mapping and mutating palmitoylation sites and comparing the activity of 'wild type' and palmitoyl-mutant nanobodies (e.g. does a palmitoyl-mutant nanobody show broader specificity against additional target proteins?)) would be useful.
- v) There are also questions regarding the overall selectivity and specificity of the system and its current reagents. The authors use flotillin as a negative control protein, but it would be informative to know the effective 'radius of depalmitoylation' of the nanobody e.g. does it depalmitoylate additional binding partners of the GFP-tagged protein of interest? It is admittedly difficult to address this issue exhaustively, but lysates from the experiment in Fig 4B could perhaps be probed for other known palmitoylated interactors of Cav1.2, if any are expressed in HEK cells. An additional control would be to perform the experiment in Fig4C-F using a palmitoyl-mutant Cav1.2 (seeing no further shift would provide evidence for specificity).
- vi) Related to (v), is the trimethoprim-bound nanobody inactive against all substrates, or can it now depalmitoylate additional cellular palmitoyl-proteins? Any information regarding this point could be an important caveat to the system.
- vii) A major current limitation of the study is that the method is only demonstrated to be effective if the target protein or its interactors are over-expressed in heterologous cells. It would be most convincing if efficacy of the system were to be demonstrated against an endogenous protein, but admittedly this is a challenging task that would likely require a new nanobody to be generated. More realistically, though, can the authors show efficacy in a system in which an endogenous palmitoyl-protein is knocked down or knocked out and then replaced by a GFP-tagged version expressed at close to endogenous levels? Such evidence would support the broader future applicability of this approach. If it were possible to address (v) in such a system that would further strengthen the study.
- viii) The trimethoprim experiment is impressive, but it would be helpful to know how quickly after trimethoprim treatment effects can be seen (this information should be provided), as well as whether this effect is reversible and, if so, what the kinetics of reversibility are. While the latter points are less critical for the current study, all such information would help others evaluate the potential usefulness of this system for their own experiments.

Minor points:

- i) Given that the current manuscript only demonstrates efficacy of the system using transiently or stably (i.e. not

endogenously) expressed proteins, the statement at the end of the abstract regarding clinical applications should likely be toned down or removed.

ii) It might be helpful to add 1-2 sentences regarding specific future applications at the end of the Discussion.

Reviewer #3

(Remarks to the Author)

In this work, Kuo et al. described a method that uses functionalized nanobodies to specifically target protein palmitoylation, or essentially depalmitoylate proteins. As has been pointed out by these authors, posttranslational protein modifications play a key role to control protein localization, interaction, and function. Therefore, this work is of value for scientists to study the mechanisms associated with palmitoylation. On the other hands, however, this piece of work involves a number of weaknesses and/or pitfalls that need to be addressed. The following suggestions are provided to improve the manuscript.

Major issues:

1. Overexpression of any enzyme within a cell is known to have harmful effects on the entire proteome. In this study, the authors focus on the overexpression of a nanobody-enzyme chimera, more specifically a thioesterase. Therefore, it is anticipated that the overexpressed thioesterase will impact the entire proteome by causing palmitoylation of a wide range of protein substrates, ultimately leading to negative consequences for the cell. Hence, the authors need to provide additional solid experimental evidences to justify this. A highly recommended experiment is quantitative proteomics analysis to demonstrate changes in protein levels and palmitoylation status of the proteome with and without overexpression of the nanobody-thioesterase chimera. Additional experiments, such as cell viability assays, may also be necessary. If the quantitative proteomics study does not show any off-target effects (which is highly unlikely), the authors must offer convincing explanations supported by experiments as to why overexpression of an enzyme chimera does not impact cell viability, especially in light of contradictory studies like PhosTAC, which is also a tool for control of PTM (i.e. dephosphorylation).

2. In relation to question 1, various strategies can be employed to minimize off-target effects in biotechnologies based on the 'induced proximity' mechanism. It is strongly advised for the authors to suggest a specific approach for addressing or mitigating potential off-target effects, and to provide a practical example. Furthermore, it is recommended to discuss alternative solutions for addressing off-target effects caused by overexpressed enzymes within a cell in the discussion section.

3. Figure 1B and other similar experiments seem to be not properly presented. A data shows the presence of both the palmitoylated and non-palmitoylated Ca(v)1.2 in the whole cell lysate could reveal the increase or decrease of palmitoylation states, but not like the data shown in Figure 1B. The results in Figure 1B may essentially means nothing, which could not be interpreted. Due to the presence of palmitoyl chain, palmitoylated or non-palmitoylated protein could show different shifts in the blots.

4. Please include a time-dependent study of the de-palmitoylation process for Figure 1B, and other similar figures.

5. Please provide original data, such as original uncropped western blot images as supplement.

6. There are necessary and important controls not being included while some abundant controls being added. For example, the data and information like Figure 1C, 1D, 1E and 1F are essentially not necessary and should be removed from the manuscript. The provided control information is simply intuitive.

7. The references in this work need to be improved. For instance, when discussing induced proximity for the control of PTM, it is essential to cite relevant works such as PMID: 34780684. Additionally, since this study relies on the mechanism of nanobody-based induced proximity for targeted de-palmitoylation of proteins, it is recommended to include recent papers like PMID: 36964170. The authors should carefully review the references to ensure that the most pertinent works are not overlooked.

8. Numerous errors were identified in the methodology section, such as the absence of catalogue numbers for key reagents, such as antibodies. Other examples of errors include 'Ba²⁺' instead of 'Ba²⁺', '1.6μg' instead of '1.6 μg', and 'MgCl₂' instead of 'MgCl₂'(subscript). The authors are advised to thoroughly review and revise the method section to correct these errors.

9. The figure captions are not complete nor fluent. Please carefully check and revise. For example, Figure 1A: Scheme. What does it mean? There is no information about it.

10. Discussion is too lengthy and not focus. Please do not repeat the results again in the Discussion but just list the key findings. And what is the advantage and potential of the reported methods, what is the limitations, and the what are potential means to address the limitations?

11. The authors mentioned in the manuscript "...but often relies on tools that impact entire pathways rather than individual proteins..." Is the description accurate? Please justify this comment and preferably add examples and cite references to support this comment. Otherwise, remove it or modify it.

12. The authors mentioned in the manuscript: "A chemogenetic approach to control nanobody affinity for its target enable

temporal control of protein palmitoylation". This is not accurate. Please specify activation or deactivation, not just vaguely "control". Again, this chemo-genetic approach does not address the potential off-target effect unless the chemogenetic approach is able to activate the de-palmitoylation process.

13. The authors mentioned in the manuscript: "with broad applications in both the laboratory and the clinic". The use of "broad" is too strong. And how is it related to clinic? As far as this reviewer knows, the reported protocol requires genetic modification (transfection). Is this ethically allowed in clinic?

14. "We reasoned that fusing the nanobody to the amino termini would interfere with, but might substitute for, this membrane targeting. Therefore, nanobody fusions to the amino terminus of the thioesterases did not include a flexible linker between the two proteins. We included a flexible linker when fusing LaG-16 to the carboxyl termini of thioesterases." This reviewer is not fully clear why a linker is needed or not needed when fuse a nanobody to the N or C terminus of an esterase. Please improve the description to make it more clearly explained.

Minor points

1. Use of stars (*) to denote statistical significance is not recommended. Please provide exact P values.
2. What is the ethics permission number?
3. The displayed figures should be thoroughly improved. For example, some of the words are too large (e.g. Fig 2), while some of the words are too small to read (e.g. Fig. 1). Please thoroughly improve all the figures and keep consistency.
4. Please use professional chemistry software, such as ChemDraw to draw the chemical structures (e.g. the palmitoylate chain in Figure 1A).
5. The authors used functionalized nanobodies to describe a nanobody-enzyme chimera, which is not accurate or correct. A functionalized nanobody usually refers to a nanobody being chemically modified to bearing an additional functionality (PMID: 28913971). It is better to use alternative terms, such as nanobody chimera, or more precisely nanobody-thioesterase chimera etc.
6. One-way ANOVA analysis. Please give the full name of this analysis (ANOVA) and possibly indicate what it is used for.
7. Are some designations/abbreviations properly or clearly used? For example, UF, and P used in Figure 1.
8. Is flot2 in the WB images act as an internal standard?
9. A nanobody contains three CDRs. Please revise the scheme of nanobody in Figure 1A.

Version 1:

Reviewer comments:

Reviewer #1

(Remarks to the Author)

The authors have done a number of new experiments that have addressed my previous concerns.

Reviewer #2

(Remarks to the Author)

The revised manuscript is significantly strengthened, and the major concerns of this reviewer are now answered. There are some minor technical/formatting points that would be good to address. In particular:

- i) The new live cell images and movies are an important addition. However, while the effect is reasonably convincing, the GFP-Spry2 signal in these experiments is quite saturated in this reviewer's copy, making it harder to appreciate the effect. If the authors have images/movies from lower expressing cells this would be very helpful (and might make the effect even more apparent). Any quantification of data from the still images and/or movies would also be helpful, but is not essential.
- ii) There is a typo (S112A) in Fig 5F
- iii) In Fig 6A (another nice experiment that further strengthens the study), the example bottom-right trace shows no EAD, but this result is apparently only rarely observed (5/21 cases i.e. approx. 25% in Fig 6C). It would be more accurate/representative to show a set of 4-5 individual traces per condition, in which only one trace should show no EAD.

Reviewer #3

(Remarks to the Author)

In the revised manuscript, the authors have made efforts to perform additional experiments attempting to address the concerns raised previously. These experiments include a proteome-wide profiling of the overexpression of nanobody-esterase chimera, among others. The authors have also added further discussions. While I appreciate these revisions, I still have some concerns and suggestions, which are listed below:

1. According to the quantitative proteomic study results, it is clear that the overexpression of an enzyme causes changes in the status of proteins, particularly the palmitoylation level. Therefore, it is necessary to perform cell viability assays, such as EdU plus CCK-8 or EdU and MTT assays using different cell types, to examine the impact of proteome palmitoylation status alternation on the proliferation and viability of live cells. It seems that the authors applied some self-defined filtration criteria to reduce the total number of proteins from 4155 to only 2869. Is this filtration perhaps a bit too strong? Additionally, identifying fewer than 3000 proteins appears to be somewhat low for a typical quantitative proteomics experiment (typically 4000 or more, and ideally 7000 or more).

2. I still do not agree with presenting a cartoon image of Cys palmitoylation without specifying the exact chemical structure. A cartoon image can mean anything, and cysteine can be post-translationally modified by different moieties, such as palmitoyl group, farnesyl group, sulfenyl, or simply oxidized. It is not only unprofessional to omit the chemical structure of the post-translational modification, but it also creates ambiguity in scientific illustration.

3. Following the above point, what is the exact molecular weight change after palmitoylation or depalmitoylation? This information should be mentioned in the manuscript, particularly in the sections related to the proteomic profiling of cysteine palmitoylation/ depalmitoylation.

4. Additional suggestions for references: In the first paragraph discussing the background of PTMs, the authors have overlooked the use of semi-synthetic post-translationally modified proteins to study the role of specific PTM processes (representative work, e.g., PMID: 20512138). This classic approach avoids experimental artifacts and does not trigger off-target effects.

REVIEWER COMMENTS

Reviewer #1 (Remarks to the Author):

The addition of palmitate to proteins is a frequent post-translational modification that affects as much as 10% of the human proteome, often with important functional consequences. However, the field lacks specific reagents to test its relevance for a given substrate, particularly in a reversible manner. In this manuscript, the authors present a strategy for selectively removing post-translational palmitate modifications from proteins, specifically the alpha subunit of the Ca(v)1.2 channel. Because palmitoylation modifies channel conductance which is dysregulated in disease, new techniques for its removal could have therapeutic implications.

The authors used the nanobody-based recruitment of a cellular depalmitoylation enzyme (APT1 or APT2) to selectively target this channel. Different strategies were first developed as a proof-of-concept. APT2 (or the related protein APT1) was coupled to a GFP-targeted nanobody, and was shown to efficiently depalmitoylate a GFP-tagged, dually-palmitoylated fragment of Cav1.2. As expected, this depalmitoylation required APT catalytic activity, and binding of the nanobody to GTP.

The ability of these nanobody fusions to target other GFP-tagged palmitoylated proteins was also tested. This uncovered substantial differences in enzyme specificity. APT1, but not APT2, was effective on the cardiac sodium/calcium exchanger NCX1. In contrast, APT2 worked well on the Cav1.2 fragment, and had a more modest effect on paralectin. The reason for this specificity is unclear, and while it could be due to subtle conformational differences that affect substrate access, it could also suggest an inherent selectivity that would be interesting to follow up in future studies.

The authors further used the ligand-modulated antibody fragment (LAMA) approach to make a chemically regulated version of their GFP-directed depalmitoylation construct. Adding a chemical to disrupt nanobody binding led to a substantial recovery of palmitoylation. This is a very useful tool, and I would be very interested to see it used to address a biological question.

Finally, to target Ca(v)1.2 in cells, they fused APT2 to a nanobody that binds the Ca(v)1.2 beta subunit. When Ca(v)1.2 alpha and beta subunits were co-expressed in HEK cells, nanobody-based targeting of APT2 to the Ca(v)1.2 beta subunit resulted in depalmitoylation of the Ca(v)1.2 alpha subunit. Importantly, voltage clamp experiments showed this resulted in a shift in the voltage dependence of activation of Ca(v)1.2 similar to that previously seen when the palmitoylation site was mutated.

Overall, I found the nanobody-targeting approach for selective protein depalmitoylation to be novel and exciting: it has the potential to address important questions about the role of palmitoylation in the cellular environment. I was however disappointed that these tools were not used to gain new biological insights in the current work. Nevertheless, the demonstration that nanobody-targeted depalmitoylases can modulate the activation of Ca(v)1.2 is significant for its potential to spur the development of new therapeutic approaches.

We recognise the reviewer's disappointment that the tools we developed were not applied to gain new insights into biology. In the revised manuscript we have use the nanobody directed against Ca(v)1.2 to depalmitoylate the channel's alpha subunit and reduce arrhythmia susceptibility in induced pluripotent stem cell derived cardiac myocytes (Fig 6). We hope the reviewer agrees that this highlights one of many potential therapeutic applications for this technology.

Specific comments:

1. Few details were provided of experiments using the chemically-controlled (LAMA-modified) nanobody. How long were cells treated with trimethoprim, and what were the kinetics of re-palmitoylation?

We have clarified that trimethoprim was typically applied overnight in the revised manuscript. We agree with the reviewer that it is important to study the kinetics. The kinetics of re-palmitoylation will vary according to the zDHHC-PAT responsible for palmitoylating a particular protein. Rather than investigate this re-palmitoylation rate (which could be different for every substrate) we have investigated how quickly the LAMA is capable of depalmitoylating its target protein when trimethoprim is withdrawn from cells, allowing the nanobody to bind its target. This turns out to be remarkably quick: 100% of the target protein is depalmitoylated within an hour of withdrawing trimethoprim. These experiments are described in lines 162-181 and presented in a new Figure 4 in the revised manuscript.

2. With the exception of the voltage clamp experiments, all experiments used acyl-RAC. This is an appropriate technique to study palmitoylation, and these studies were well-conducted and appropriately interpreted. However, acyl-RAC reports on all sites, even those not normally subjected to dynamic turnover. It was not clear to me if the nanobody constructs only act on the subset of dynamic sites, or also targets static modifications. This could be addressed using other approaches (ie. metabolic labeling with click-able analogs) and additional substrates. The paper could also benefit from use of additional complementary techniques, such as the use of imaging to show if the nanobody caused localization changes for any of the substrates studied.

Thank you for these suggestions. We have undertaken several experiments to address these points:

- New Figure 2C (described in lines 134-142) evaluating whether nanobodies can depalmitoylate the constitutively palmitoylated protein caveolin-3.
- New cell fractionation experiments to evaluate nanobody ability to relocalise a substrate to the cytosol (Figure 4C, described in lines 172-181).
- New imaging experiments to visualise localisation changes induced by nanobody-induced depalmitoylation (Figure 4D, Supplementary Movies 1-3, described in lines 172-181)

Reviewer #2 (Remarks to the Author):

Kuo et al. describe the development and functional characterization of intracellularly expressed nanobodies fused to protein depalmitoylases, which they show can depalmitoylate specific transiently co-expressed or stably expressed substrates in cell lines. In the latter case, involving a specific calcium channel subunit, the authors report a functional change in voltage-dependence of the channel. The authors also demonstrate that such depalmitoylation can be temporally controlled by generating a trimethoprim-sensitive version of the nanobody-depalmitoylase fusion protein.

This is an interesting manuscript that has considerable potential relevance not just for the field of protein lipidation, but also for other areas of post-translational regulation. However, the current findings are slightly limited, and the impact of the study would be increased by demonstrating the use of this approach in a more physiological context and/or by providing more evidence of its selectivity and/or advantages compared to similar methods. It is not essential that the authors address all of the issues below (and this reviewer hesitates to request numerous additional experiments), but a more in-depth characterization would make the potential benefits of this system easier to appreciate, particularly for the broad audience that could potentially use this methodology in the future. In particular:

i) It would be helpful if the authors were to experimentally benchmark their system against other, similar tunable approaches (most obviously rapalog-based dimerization systems). One obvious advantage is that the GFP tag can potentially report a change in localization of the target protein after depalmitoylation in fixed and even live cells. Such a demonstration, particularly for the trimethoprim-inducible version of the nanobody (and perhaps more readily shown for a cytosolic palmitoyl-protein rather than a channel) would greatly strengthen the manuscript and would provide evidence that the method can provide insights that rapalog-based systems cannot.

Thank you for this suggestion. We have used cell fractionation and imaging to evaluate changes in localisation of eGFP-Sprouty2 upon trimethoprim withdrawal to 'activate' nanobody binding. These highlight the utility of the system. See Figure 4, Supplementary Movies 1-3, described in lines 162-181.

ii) Conversely, one of the main advantages of this system (which is also potentially possible with rapalog-binding fusion protein systems) is that a protein of interest can be palmitoylated in cells and then the functional impact of its depalmitoylation can be assessed, potentially on-demand. In contrast, most prior methods rely on a mutant (usually C->A or C->S) that is never palmitoylated in the first place. Surprisingly, the authors do not emphasize this point but may wish to make it more forcefully in the Introduction and/or Discussion

We highlight that this point was already mentioned in the introduction (lines 52-56). We now re-emphasise the point in the discussion (lines 263-272) and provide comparisons to the rapalog-based systems relevant to reviewer 2 point (i).

iii) With (ii) in mind, though, the authors should acknowledge (and cite accordingly) that a similar system was recently developed for protein phosphatases (PMID: 36720221). The current study provides support for the broader applicability of this method, but the statement in the Discussion about the approach being limited to ubiquitination should be modified. As with (i), if the authors have introduced specific enhancements beyond those reported for phosphatase- and/or E3 ligase-based systems these could be highlighted.

We are very grateful to the reviewer for pointing this out. We were not aware of the study, and we have developed a very similar system in the lab already. We have edited the discussion (lines 263-272) to highlight this work and contextualise our system with this and other systems.

iv) It is unclear if the nanobody itself becomes palmitoylated (apparently yes, in Fig 4B) and how this might affect its efficacy and selectivity. An experimental demonstration of this point (by mapping and mutating palmitoylation sites and comparing the activity of 'wild type' and palmitoyl-mutant

nanobodies (e.g. does a palmitoyl-mutant nanobody show broader specificity against additional target proteins?) would be useful.

We apologise for not covering this point in the original manuscript. The nanobody thioesterase fusions are only palmitoylated when the nanobody is fused at the C terminus. Palmitoylation is already established to occur at the N terminus of the APTs, and we find that this palmitoylation is prevented when a nanobody is fused here. Palmitoylated nanobodies (ie the C terminal fusion) were marginally more potent than non-palmitoylated nanobodies. Importantly neither reagent depalmitoylates the housekeeping protein flot2 (Figure 1). These points are now highlighted in the results (lines 99-100) and in Figure 1B.

v) There are also questions regarding the overall selectivity and specificity of the system and its current reagents. The authors use flotillin as a negative control protein, but it would be informative to know the effective 'radius of depalmitoylation' of the nanobody e.g. does it depalmitoylate additional binding partners of the GFP-tagged protein of interest? It is admittedly difficult to address this issue exhaustively, but lysates from the experiment in Fig 4B could perhaps be probed for other known palmitoylated interactors of Cav1.2, if any are expressed in HEK cells. An additional control would be to perform the experiment in Fig4C-F using a palmitoyl-mutant Cav1.2 (seeing no further shift would provide evidence for specificity).

These are excellent points. We note in the discussion (line 251) that depalmitoylation of Ca(v)1.2 alpha subunit can be achieved using a nanobody that binds the beta subunit. In other words, the effective radius of depalmitoylation undoubtedly stretches beyond only the nanobody's target protein. We suggest that future studies will address this question comprehensively if they employ nanobodies that bind proteins of interest rather than tags. Ca(v)1.2 is probably not the best target to evaluate this because although one isoform of its beta subunit is palmitoylated (beta2a – a non-cardiac subunit), this is widely believed to be a stably rather than dynamically palmitoylated protein.

vi) Related to (v), is the trimethoprim-bound nanobody inactive against all substrates, or can it now depalmitoylate additional cellular palmitoyl-proteins? Any information regarding this point could be an important caveat to the system.

Thank you for this suggestion. We have comprehensively evaluated off-target effects of this nanobody using palmitoyl proteomics (described in lines 182-200, Figure 4E and Supplementary Figures 2-4). We also highlight in the discussion a future direction to further improve the specificity of the system by engineering TMP sensitivity into the enzyme as well as the nanobody (lines 277-281).

vii) A major current limitation of the study is that the method is only demonstrated to be effective if the target protein or its interactors are over-expressed in heterologous cells. It would be most convincing if efficacy of the system were to be demonstrated against an endogenous protein, but admittedly this is a challenging task that would likely require a new nanobody to be generated. More realistically, though, can the authors show efficacy in a system in which an endogenous palmitoyl-protein is knocked down or knocked out and then replaced by a GFP-tagged version expressed at close to endogenous levels? Such evidence would support the broader future applicability of this approach. If it were possible to address (v) in such a system that would further strengthen the study.

Thank you for this suggestion. We have extended the approach to target an endogenous protein: Ca(v)1.2 in induced pluripotent stem cell derived cardiac myocytes (iPSC-CMs). Extending the experiments originally presented in Figure 4 (Figure 5 in the revised manuscript - demonstrating Ca(v)1.2 nanobody-mediated changes in voltage dependence of this channel in engineered cell lines), we now demonstrate that by changing the voltage dependence of the channel we reduce the susceptibility of iPSC-CMs to arrhythmias in a setting of action potential prolongation (described in lines 225-242, Figure 6).

viii) The trimethoprim experiment is impressive, but it would be helpful to know how quickly after trimethoprim treatment effects can be seen (this information should be provided), as well as whether

this effect is reversible and, if so, what the kinetics of reversibility are. While the latter points are less critical for the current study, all such information would help others evaluate the potential usefulness of this system for their own experiments.

We agree. Typically, trimethoprim was applied overnight. In a new series of experiments, we have investigated how quickly the LAMA is capable of depalmitoylating its target protein when trimethoprim is withdrawn from cells, allowing the nanobody to bind its target. This turns out to be remarkably quick: 100% of the target protein is depalmitoylated within an hour of withdrawing trimethoprim. These experiments are described in lines 172-181 and presented in a new Figure 4 and Supplementary Movies 1-3 in the revised manuscript.

Minor points:

- i) Given that the current manuscript only demonstrates efficacy of the system using transiently or stably (i.e. not endogenously) expressed proteins, the statement at the end of the abstract regarding clinical applications should likely be toned down or removed.
- ii) It might be helpful to add 1-2 sentences regarding specific future applications at the end of the Discussion.

Thank you. Since we have gone on to show changes in arrhythmia sensitivity in iPSC-CMs we have left this statement in the abstract. Future applications of the technology are now discussed (lines 260-262 & 308-310).

Reviewer #3 (Remarks to the Author):

In this work, Kuo et al. described a method that uses functionalized nanobodies to specifically target protein palmitoylation, or essentially depalmitoylate proteins. As has been pointed out by these authors, posttranslational protein modifications play a key role to control protein localization, interaction, and function. Therefore, this work is of value for scientists to study the mechanisms associated with palmitoylation. On the other hands, however, this piece of work involves a number of weaknesses and/or pitfalls that need to be addressed. The following suggestions are provided to improve the manuscript.

Major issues:

1. Overexpression of any enzyme within a cell is known to have harmful effects on the entire proteome. In this study, the authors focus on the overexpression of a nanobody-enzyme chimera, more specifically a thioesterase. Therefore, it is anticipated that the overexpressed thioesterase will impact the entire proteome by causing palmitoylation of a wide range of protein substrates, ultimately leading to negative consequences for the cell. Hence, the authors need to provide additional solid experimental evidences to justify this. A highly recommended experiment is quantitative proteomics analysis to demonstrate changes in protein levels and palmitoylation status of the proteome with and without overexpression of the nanobody-thioesterase chimera. Additional experiments, such as cell viability assays, may also be necessary. If the quantitative proteomics study does not show any off-target effects (which is highly unlikely), the authors must offer convincing explanations supported by experiments as to why overexpression of an enzyme chimera does not impact cell viability, especially in light of contradictory studies like PhosTAC, which is also a tool for control of PTM (i.e. dephosphorylation).

Thank you for this suggestion. We have conducted a full characterisation of whole cell proteome and palmitoyl proteome in cells engineered to stably express tetracycline inducible APT2-LAMA-G97. (described in lines 182-200, Figure 4E and Supplementary Figures 2-4). We find that a small fraction of the cellular palmitome (~5%) is remodelled when the chimera is expressed, but the whole cell proteome is essentially unchanged. Hence there are indeed limited off-target effects of these reagents. We acknowledge this limitation, and also highlight in the discussion a future direction to further improve the specificity of the system by engineering TMP sensitivity into the depalmitoylating enzyme as well as the nanobody (lines 278-281).

2. In relation to question 1, various strategies can be employed to minimize off-target effects in biotechnologies based on the 'induced proximity' mechanism. It is strongly advised for the authors to suggest a specific approach for addressing or mitigating potential off-target effects, and to provide a practical example. Furthermore, it is recommended to discuss alternative solutions for addressing off-target effects caused by overexpressed enzymes within a cell in the discussion section.

Thank you for this suggestion. We have added a section to the discussion addressing this point. We hypothesise that it should be possible to improve the specificity of the system by engineering TMP sensitivity into the enzyme as well as the nanobody (lines 278-281). This will be the subject of future investigations.

3. Figure 1B and other similar experiments seem to be not properly presented. A data shows the presence of both the palmitoylated and non-palmitoylated Ca(v)1.2 in the whole cell lysate could reveal the increase or decrease of palmitoylation states, but not like the data shown in Figure 1B. The results in Figure 1B may essentially means nothing, which could not be interpreted. Due to the presence of palmitoyl chain, palmitoylated or non-palmitoylated protein could show different shifts in the blots.

The assay used in Figure 1B and throughout the paper is resin-assisted capture of acylated proteins (acyl-RAC). Here we assess abundance of a protein of interest (UF: unfractionated cell lysate) alongside the purified palmitoylated fraction (P: proteins captured in the acyl-RAC assay). We measure band intensity of each and express the intensity of 'P' relative to the intensity of the

corresponding 'UF' (ie amount captured / total abundance). This assay has been widely used for many years in the field.

It is rare for a protein to show different mobility on SDS PAGE solely as a result of being palmitoylated (although examples exist for heavily palmitoylated proteins such as cysteine string protein – PMID 16943324). Hence, we rely on acyl-RAC rather than electrophoretic mobility to define the fraction of a protein that is palmitoylated. Acyl-RAC does not directly tell us the amount of non-palmitoylated protein in a cell lysate. This can be inferred by comparing 'capture' (P) to 'abundance' (UF), but we prefer to present 'the fraction of the protein that is captured' (ie what is palmitoylated) rather than the less direct 'the fraction of the protein that is not captured' (ie what is non-palmitoylated).

4. Please include a time-dependent study of the de-palmitoylation process for Figure 1B, and other similar figures.

The experiment presented in Figure 1B is a steady state snapshot of palmitoylation levels of a protein expressed 18-24 hours after transient transfection or 18-24 hours after it was transiently co-transfected with a nanobody. A time-dependent study would show that these proteins gradually increase in expression after transfection. We have elected to present instead the time dependence of depalmitoylation when the trimethoprim 'brake' is removed from the LAMAs to allow them to bind and depalmitoylate their substrates. This turns out to be remarkably quick: 100% of the target protein is depalmitoylated within an hour of withdrawing trimethoprim. These experiments are described in lines 172-181 and presented in a new Figure 4 and Supplementary Movies 1-3 in the revised manuscript.

5. Please provide original data, such as original uncropped western blot images as supplement.

These are provided as a supplementary file with the revised manuscript.

6. There are necessary and important controls not being included while some abundant controls being added. For example, the data and information like Figure 1C, 1D, 1E and 1F are essentially not necessary and should be removed from the manuscript. The provided control information is simply intuitive.

We have discussed this suggestion with the editor. We respectfully suggest that it is important to present these controls. The possibility that nanobody binding blocks access for a palmitoylating enzyme is real and should be tested. To ignore these controls and argue that intuitively the results agree with our expectations is a road to disaster.

7. The references in this work need to be improved. For instance, when discussing induced proximity for the control of PTM, it is essential to cite relevant works such as PMID: 34780684. Additionally, since this study relies on the mechanism of nanobody-based induced proximity for targeted de-palmitoylation of proteins, it is recommended to include recent papers like PMID: 36964170. The authors should carefully review the references to ensure that the most pertinent works are not overlooked.

Thank you for these suggestions. We have discussed the papers the reviewer suggests (discussion lines 263-281).

8. Numerous errors were identified in the methodology section, such as the absence of catalogue numbers for key reagents, such as antibodies. Other examples of errors include 'Ba2+' instead of 'Ba²⁺', '1.6µg' instead of '1.6 µg', and 'MgCl2' instead of 'MgCl₂'(subscript). The authors are advised to thoroughly review and revise the method section to correct these errors.

We refer the reviewer to the reporting summary that accompanies the manuscript, which includes catalogue and lot numbers for all antibodies used in the investigation. We apologise for errors in the methods, which we hope the reviewer will agree we have corrected in the revision.

9. The figure captions are not complete nor fluent. Please carefully check and revise. For example, Figure 1A: Scheme. What does it mean? There is no information about it.

Apologies. We have revised legends to improve clarity.

10. Discussion is too lengthy and not focus. Please do not repeat the results again in the Discussion but just list the key findings. And what is the advantage and potential of the reported methods, what is the limitations, and the what are potential means to address the limitations?

Thank you. We have revised the discussion to improve the focus and address the suggested points. We cover the following points in the discussion:

1. Contextualise our use of nanobodies to target palmitoylation with other examples targeting PTMs in the literature.
2. Discuss our chemogenetic approach in the context of other induced proximity systems, highlighting advantages and limitations, with suggestions for future work to address these limitations.
3. Discuss the substrate-specificity of our reagents, highlighting the need for future work to address how thioesterase enzymes identify their individual substrates.
4. Discuss the importance of nanobody affinity for target depalmitoylation, highlighting the need for future work to identify the affinity range across which these reagents are effective.

11. The authors mentioned in the manuscript "...but often relies on tools that impact entire pathways rather than individual proteins..." Is the description accurate? Please justify this comment and preferably add examples and cite references to support this comment. Otherwise, remove it or modify it.

Here we are referring to inhibitors that block a particular signalling pathway. We have removed this point from the abstract.

12. The authors mentioned in the manuscript: "A chemogenetic approach to control nanobody affinity for its target enable temporal control of protein palmitoylation". This is not accurate. Please specify activation or deactivation, not just vaguely "control". Again, this chemo-genetic approach does not address the potential off-target effect unless the chemogenetic approach is able to activate the de-palmitoylation process.

We have revised this statement to improve clarity and highlight that the chemogenetic approach enhances depalmitoylation of the target (line 24).

13. The authors mentioned in the manuscript: "with broad applications in both the laboratory and the clinic". The use of "broad" is too strong. And how is it related to clinic? As far as this reviewer knows, the reported protocol requires genetic modification (transfection). Is this ethically allowed in clinic?

We highlight that several trials of gene therapy have been conducted in humans (in setting of heart failure, the CUPID-1 phase 2 and CUPID-2 phase 2b trials are often cited). It is reasonable to propose using a genetic instrument to treat human disease.

14. "We reasoned that fusing the nanobody to the amino termini would interfere with, but might substitute for, this membrane targeting. Therefore, nanobody fusions to the amino terminus of the thioesterases did not include a flexible linker between the two proteins. We included a flexible linker when fusing LaG-16 to the carboxyl termini of thioesterases." This reviewer is not fully clear why a linker is needed or not needed when fuse a nanobody to the N or C terminus of an esterase. Please improve the description to make it more clearly explained.

Apologies for our lack of clarity here. We have amended the statement (lines 84-86).

Minor points

1. Use of stars (*) to denote statistical significance is not recommended. Please provide exact P values.

Corrected.

2. What is the ethics permission number?

This is now provided

3. The displayed figures should be thoroughly improved. For example, some of the words are too large (e.g. Fig 2), while some of the words are too small to read (e.g. Fig. 1). Please thoroughly improve all the figures and keep consistency.

Apologies. We have adjusted these font sizes.

4. Please use professional chemistry software, such as ChemDraw to draw the chemical structures (e.g. the palmitoylate chain in Figure 1A).

Figure 1A is now presented in cartoon format which we suggest would not be enhanced by using ChemDraw to depict the palmitate.

5. The authors used functionalized nanobodies to describe a nanobody-enzyme chimera, which is not accurate or correct. A functionalized nanobody usually refers to a nanobody being chemically modified to bearing an additional functionality (PMID: 28913971). It is better to use alternative terms, such as nanobody chimera, or more precisely nanobody-thioesterase chimera etc.

Corrected.

6. One-way ANOVA analysis. Please give the full name of this analysis (ANOVA) and possibly indicate what it is used for.

Corrected.

7. Are some designations/abbreviations properly or clearly used? For example, UF, and P used in Figure 1.

We provide a full list of abbreviations used in the manuscript (line 35).

8. Is flot2 in the WB images act as an internal standard?

Yes. This is stated in the description of Figure 1 (line 97).

9. A nanobody contains three CDRs. Please revise the scheme of nanobody in Figure 1A.

Corrected.

Reviewer #2 (Remarks to the Author):

The revised manuscript is significantly strengthened, and the major concerns of this reviewer are now answered. There are some minor technical/formatting points that would be good to address. In particular:

i) The new live cell images and movies are an important addition. However, while the effect is reasonably convincing, the GFP-Spry2 signal in these experiments is quite saturated in this reviewer's copy, making it harder to appreciate the effect. If the authors have images/movies from lower expressing cells this would be very helpful (and might make the effect even more apparent). Any quantification of data from the still images and/or movies would also be helpful, but is not essential.

The issue of saturation in the images is almost unavoidable. We tried our best to eliminate it, but this comes from a technical problem. The chemogenetic nanobody is based on the GFP enhancer nanobody. Consequently, when we withdraw trimethoprim to allow the nanobody to engage GFP, the binding of the nanobody increases the fluorescent intensity of the GFP. It is the increase in fluorescence caused by the nanobody binding that is causing the images to appear saturated. To avoid the saturation, we need to visualise cells expressing less of the GFP-Spry2, but then the reader can barely see the distribution of the protein before the nanobody is 'activated' by trimethoprim withdrawal.

In the revised manuscript we have explained this limitation in the legend to the supplementary movies.

ii) There is a typo (S112A) in Fig 5F

Thank you for spotting this. We have corrected the figure.

iii) In Fig 6A (another nice experiment that further strengthens the study), the example bottom-right trace shows no EAD, but this result is apparently only rarely observed (5/21 cases i.e. approx. 25% in Fig 6C). It would be more accurate/representative to show a set of 4-5 individual traces per condition, in which only one trace should show no EAD.

Thank you for this excellent suggestion. We have redrawn Figure 6 with several example traces from each condition.

Reviewer #3 (Remarks to the Author):

In the revised manuscript, the authors have made efforts to perform additional experiments attempting to address the concerns raised previously. These experiments include a proteome-wide profiling of the overexpression of nanobody-esterase chimera, among others. The authors have also added further discussions. While I appreciate these revisions, I still have some concerns and suggestions, which are listed below:

1. According to the quantitative proteomic study results, it is clear that the overexpression of an enzyme causes changes in the status of proteins, particularly the palmitoylation level. Therefore, it is necessary to perform cell viability assays, such as EdU plus CCK-8 or EdU and MTT assays using different cell types, to examine the impact of proteome palmitoylation status alternation on the proliferation and viability of live cells. It seems that the authors applied some self-defined filtration criteria to reduce the total number of proteins from 4155 to only 2869. Is this filtration perhaps a bit too strong? Additionally, identifying fewer than 3000 proteins appears to be somewhat low for a typical quantitative proteomics experiment (typically 4000 or more, and ideally 7000 or more).

Our whole proteome experiments were filtered using standard approaches. Cellular stress typically manifests as dynamic and co-ordinated remodelling of expression of several hundred

proteins (for example PMID: 38012130). Given that our quantitative proteomics indicates only one protein has changed abundance in cells stably expressing nanobodies, we regard it as exceptionally unlikely that the cell viability is impacted. In the revised manuscript we provide evidence that stem cell derived myocytes are not adversely affected by transfection with nanobodies fused to active versus inactive APT2 (Supplementary Figure 5). We acknowledge the possibility that expressing active enzymes fused to nanobodies could conceivably impact cell viability in the discussion (lines 289-293).

2. I still do not agree with presenting a cartoon image of Cys palmitoylation without specifying the exact chemical structure. A cartoon image can mean anything, and cysteine can be post-translationally modified by different moieties, such as palmitoyl group, farnesyl group, sulfenyl, or simply oxidized. It is not only unprofessional to omit the chemical structure of the post-translational modification, but it also creates ambiguity in scientific illustration.

We have discussed this suggestion with the editor who has advised us that a schematic illustration of the palmitate is acceptable when the figure is designed only to convey the concept. We have amended the legend to Fig 1a to clarify what the schematic depicts.

3. Following the above point, what is the exact molecular weight change after palmitoylation or depalmitoylation? This information should be mentioned in the manuscript, particularly in the sections related to the proteomic profiling of cysteine palmitoylation/ depalmitoylation.

Thank you for this suggestion. We have added this information to the Results section (lines 82-84).

4. Additional suggestions for references: In the first paragraph discussing the background of PTMs, the authors have overlooked the use of semi-synthetic post-translationally modified proteins to study the role of specific PTM processes (representative work, e.g., PMID: 20512138). This classic approach avoids experimental artifacts and does not trigger off-target effects.

Thank you for this suggestion. We have included this information in an expanded Introduction (lines 57-64).